# Stochastic Composite Mirror Descent: Optimal Bounds with High Probabilities

**Yunwen Lei** and **Ke Tang**[*]

Shenzhen Key Laboratory of Computational Intelligence, Department of Computer Science and Engineering, Southern University of Science and Technology, Shenzhen 518055, China
leiyw@sustc.edu.cn tangk3@sustc.edu.cn

## Abstract

We study stochastic composite mirror descent, a class of scalable algorithms able to exploit the geometry and composite structure of a problem. We consider both convex and strongly convex objectives with non-smooth loss functions, for each of which we establish high-probability convergence rates optimal up to a logarithmic factor. We apply the derived computational error bounds to study the generalization performance of multi-pass stochastic gradient descent (SGD) in a non-parametric setting. Our high-probability generalization bounds enjoy a *logarithmical* dependency on the number of passes provided that the step size sequence is square-summable, which improves the existing bounds in expectation with a *polynomial* dependency and therefore gives a strong justification on the ability of multi-pass SGD to overcome overfitting. Our analysis removes boundedness assumptions on subgradients often imposed in the literature. Numerical results are reported to support our theoretical findings.

## 1   Introduction

Stochastic gradient descent (SGD) has found wide applications in machine learning problems due to its simplicity in implementation, low memory requirement and low computational complexity per iteration, as well as good practical behavior [2, 6, 28, 32, 41]. As an iterative method, SGD minimizes empirical errors by moving iterates along the direction of a negative gradient calculated based on a loss function on a single training example or a batch of few examples. This strategy of processing few examples per iteration makes SGD particularly suitable for large scale applications with very large data points [2, 41], which are becoming ubiquitous in the big data era.

Stochastic composite mirror descent (SCMD) is a powerful extension of SGD based on two motivations [12]. Firstly, it relaxes the Hilbert space structure of SGD by using a mirror map to capture geometric properties of data from a Banach space [4, 25]. Secondly, it exploits the problem structure by separating, at every iteration, a data-fitting term and a regularization term in structured optimization problems to obtain a desired regularization effect, which arise naturally since a regularizer is often introduced to either avoid overfitting or impose a priori information [12, 37].

Although much theoretical analysis has been performed to understand the practical behavior of SGD and SCMD, the existing theoretical results are still not quite satisfactory. Firstly, most of the existing theoretical results are stated in expectation which inevitably ignore some information on high-order moments of the random variable we are interested in. In practice, we may be more interested in high-probability bounds to understand the variability of the learned model which is also an important factor we should take into account when measuring the quality of models [32]. Secondly, the existing generalization bounds, stated in expectation, for SGD either are suboptimal

---

[*]Corresponding author

or require to impose a smoothness assumption on loss functions [13, 21]. Thirdly, a non-trivial assumption on the boundedness of subgradients is often imposed in the literature to proceed with the analysis [11, 12, 28, 32], especially in the derivation of high-probability bounds. However, this boundedness assumption may not hold if the optimization is conducted in an unbounded domain, under which scenario the derived bounds may not be intuitive.

In this paper, we aim to contribute towards a refined analysis on both convergence rates and generalization properties of SCMD. We consider both general convex and strongly convex objectives, for each of which we show that SCMD can achieve almost optimal convergence rates with high probability, which match the minimax lower rates for stochastic approximation up to a logarithmic factor [1, 25]. In particular, we identify a constraint on step sizes to guarantee the boundedness of iterates with high probability (up to a logarithmic factor). Furthermore, we apply these convergence rates related to computational errors to establish high-probability generalization bounds for the model trained by SGD through *multiple passes* over the training examples, which is a typical way of using SGD to process large datasets [20]. Our generalization bounds do not require to impose smoothness assumptions on loss functions and can be optimal up to a logarithmic factor. Surprisingly, we show that estimation errors scale logarithmically with respect to (w.r.t.) the number of passes provided that the step size sequence is square-summable, which implies that SGD may be immune to overfitting. As a contrast, estimation error bounds based on stability arguments [13] and uniform deviation arguments [21] scale polynomially w.r.t. the number of passes, which may not justify well the ability of SGD in overcoming overfitting in practice. All our theoretical results are derived without any boundedness assumptions on subgradients based on two tricks. The first trick is to use a self-bounding property of loss functions (Assumption 1) to show that a (weighted) summation of function values can be controlled by step sizes (Lemma 2). The second trick is to show that conditional variances of martingales in a one-step progress inequality of SCMD can be partially offset by some other terms in the one-step progress inequality.

The paper is organized as follows. We introduce SCMD and state convergence rates in Section 2 and Section 3, respectively. We study generalization bounds of SGD in Section 4. Discussions are given in Section 5. Simulation results and conclusions are given in Section 6 and Section 7, respectively.

## 2 Stochastic Composite Mirror Descent

Many machine learning problems involve optimization problems of a composite structure [12, 37]

$$\min_{w \in \mathcal{W}} \phi(w) = \mathbb{E}_z[f(w, z)] + r(w), \tag{2.1}$$

where $\mathcal{W}$ is a Banach space with a norm $\| \cdot \|$, $F(w) := \mathbb{E}_z[f(w, z)]$ is a data-fitting term and $r : \mathcal{W} \to \mathbb{R}_+$ is a simple regularizer possibly inducing sparsity. Here $f : \mathcal{W} \times \mathcal{Z} \mapsto \mathbb{R}_+$ is a function with $f(w, z)$ measuring the quality of a model indexed by $w \in \mathcal{W}$ on a random example $z = (x, y)$ drawn from a probability measure $\tilde{\rho}$ defined in a sample space $\mathcal{Z} = \mathcal{X} \times \mathcal{Y}$ with an input space $\mathcal{X} \subset \mathcal{W}^*$ and an output space $\mathcal{Y} \subset \mathbb{R}$. We denote by $\mathbb{E}_z$ the expectation w.r.t. $z$, and by $\mathcal{W}^*$ the dual of $\mathcal{W}$ with the dual norm $\| \cdot \|_*$. A typical choice of the data-fitting term takes the form $f(w, z) = \ell(\langle w, x \rangle, y)$, where $\ell : \mathbb{R} \times \mathcal{Y} \mapsto \mathbb{R}_+$ is a loss function and $\langle w, x \rangle$ is the dual element $x \in \mathcal{W}^*$ acting on $w \in \mathcal{W}$. With specific instantiations of loss functions $\ell$ and regularizers $r$, the formulation (2.1) covers many famous machine learning problems in a unifying framework, including least squares, support vector machines, logistic regression, lasso and elastic-net, etc [12, 37].

As an extension of SGD, SCMD uses a strongly convex and Fréchet differentiable mirror map $\Psi$ to generate an appropriate Bregman distance $D_\Psi(w, \tilde{w}) := \Psi(w) - \Psi(\tilde{w}) - \langle w - \tilde{w}, \nabla \Psi(\tilde{w}) \rangle$ to capture the involved non-Euclidean geometry [4, 25], where $\nabla \Psi(\tilde{w})$ denotes the gradient of $\Psi$ at $\tilde{w}$. Let $w_1 = 0 \in \mathcal{W}$ and $\{\eta_t\}_{t \in \mathbb{N}}$ be a positive step size sequence. Upon the arrival of $z_t$ at the $t$-th iteration, SCMD calculates a subgradient $f'(w_t, z_t) \in \partial_w f(w_t, z_t)$ as an unbiased estimate of $F'(w_t) \in \partial F(w_t)$, and updates the model as follows

$$w_{t+1} = \arg \min_{w \in \mathcal{W}} \eta_t \big[\langle w - w_t, f'(w_t, z_t) \rangle + r(w)\big] + D_\Psi(w, w_t). \tag{2.2}$$

Here $\partial_w f(w_t, z_t) := \big\{ g : f(w, z_t) - f(w_t, z_t) \geq \langle w - w_t, g \rangle \text{ for all } w \big\}$ denotes the subdifferential of $f(\cdot, z_t)$ at $w_t$. Intuitively, SCMD uses $f'(w_t, z_t)$ to form a first-order approximation of $f(\cdot, z_t)$ at $w_t$ and uses the Bregman distance $D_\Psi(w, w_t)$ to keep $w_{t+1}$ not far away from the current iterate. The regularizer $r$ is kept intact here for a regularization effect [12, 37]. A typical choice of $\Psi$ is the

$p$-norm divergence $\Psi_p(w) = \frac{1}{2}\|w\|_p^2$ ($1 < p \leq 2$), which works favorably for sparse problems by setting $p$ close to 1 [12, 37]. Here $\|\cdot\|_p$ is the $p$-norm defined by $\|w\|_p = \left(\sum_{i=1}^d |w(i)|^p\right)^{1/p}$ for $w = (w(1), \ldots, w(d)) \in \mathbb{R}^d$. SCMD recovers SGD by taking $\Psi = \Psi_2$ and $r(w) = 0$, stochastic forward-backward splitting by taking $\Psi = \Psi_2$ [11], stochastic mirror descent by taking $r(w) = 0$ [24] and stochastic mirror descent algorithm made sparse by taking $\Psi = \Psi_p$ and $r(w) = \lambda\|w\|_1$ [30].

# 3  Convergence Rates

Before stating our high-probability convergence rates, we introduce some assumptions. Throughout the paper, we assume that the mirror map $\Psi$ is Fréchet differentiable and $\sigma_\Psi$-strongly convex in the sense that $D_\Psi(w, \tilde{w}) \geq 2^{-1}\sigma_\Psi\|w - \tilde{w}\|^2$ for all $w, \tilde{w} \in \mathcal{W} \subset \mathbb{R}^d$ ($\sigma_\Psi > 0$), and $f(w, z)$ is convex w.r.t. the first argument. We also always assume that Assumption 1 and Assumption 2 hold, the sample space $\mathcal{Z}$ is bounded and $\sup_{z \in \mathcal{Z}} f(0, z) < \infty$.

**Assumption 1.** We assume that there exist $A$ and $B \geq 0$ such that the following inequalities hold for any $w \in \mathcal{W}, z \in \mathcal{Z}$ and any $f'(w, z) \in \partial f(w, z), r'(w) \in \partial r(w)$

$$\|f'(w, z)\|_*^2 \leq Af(w, z) + B \quad \text{and} \quad \|r'(w)\|_*^2 \leq Ar(w) + B. \tag{3.1}$$

This is a standard assumption and satisfied in many practical problems [11, 41]. For example, Lemma A.5 shows that $r(w) = \lambda\|w\|_p^p$ satisfies the second inequality of (3.1) with $\|\cdot\| = \|\cdot\|_p (1 \leq p \leq 2)$, $A = 2\lambda p(p-1)$ and $B = \lambda p(2-p)$. Furthermore, if $f(w, z) = \ell(\langle w, x\rangle, y)$, then Lemma A.4 shows that $\|f'(w, z)\|_*^2 = |\ell'(\langle w, x\rangle, y)|^2\|x\|_*^2$ would satisfy the first inequality of (3.1) if

$$|\ell'(a, y)|^2 \leq \tilde{A}\ell(a, y) + \tilde{B}, \quad \forall a \in \mathbb{R}, y \in \mathcal{Y} \tag{3.2}$$

for some $\tilde{A}, \tilde{B} > 0$ [41], where $\ell'(a, y)$ denotes a subgradient of $\ell$ w.r.t. the first argument. Many popular loss functions satisfy (3.2), including the $p$-norm hinge loss $\ell(a, y) = \max\{0, 1 - ya\}^p$ ($1 \leq p \leq 2$) [34], the logistic loss $\ell(a, y) = \log(1 + \exp(-ya))$ for classification, and the $p$-th power absolute distance loss $\ell(a, y) = |a - y|^p$ ($1 \leq p \leq 2$), the Huber loss $\ell(a, y) = (a - y)^2$ if $|a - y| \leq 1$ and $\ell(a, y) = 2|a - y| - 1$ otherwise for regression [41]. We refer the interested readers to [41] for constants $\tilde{A}, \tilde{B}$ in (3.2) with different loss functions $\ell$.

**Assumption 2.** We assume the existence of $\sigma_F, \sigma_r \geq 0$ such that

$$\begin{aligned} F(w) - F(\tilde{w}) - \langle w - \tilde{w}, F'(\tilde{w})\rangle &\geq \sigma_F D_\Psi(w, \tilde{w}), \\ r(w) - r(\tilde{w}) - \langle w - \tilde{w}, r'(\tilde{w})\rangle &\geq \sigma_r D_\Psi(w, \tilde{w}) \end{aligned} \tag{3.3}$$

hold for all $w, \tilde{w} \in \mathcal{W}$ and any $F'(\tilde{w}) \in \partial F(\tilde{w}), r'(\tilde{w}) \in \partial r(\tilde{w})$.

The case $\sigma_\phi := \sigma_F + \sigma_r = 0$ corresponds to general convex objectives, while the case $\sigma_\phi > 0$ corresponds to strongly convex objectives. Let $w^* = \arg\min_{w \in \mathcal{W}} \phi(w)$ be the minimizer of $\phi$ in $\mathcal{W}$ with the minimal norm. We always assume $\|w^*\| < \infty$ in this paper.

Our theoretical analysis is based on the following lemma quantifying the one-step progress of SCMD measured by Bregman distance, which shows how $D_\Psi(w, w_t)$ would change in a single iteration.

**Lemma 1.** Let $\{w_t\}_{t \in \mathbb{N}}$ be generated by (2.2), then the following inequality holds for any $w \in \mathcal{W}$

$$\begin{aligned} D_\Psi(w, w_{t+1}) - D_\Psi(w, w_t) \leq &\; \eta_t\langle w - w_t, f'(w_t, z_t)\rangle + \eta_t(r(w) - r(w_t)) \\ &+ \sigma_\Psi^{-1}\eta_t^2 \underbrace{\left(Af(w_t, z_t) + Ar(w_t) + 2B\right)}_{:=\mathfrak{A}_t} - \sigma_r\eta_t D_\Psi(w, w_{t+1}). \end{aligned} \tag{3.4}$$

Existing one-step progress inequality can be found in the literature with $\mathfrak{A}_t$ replaced by $\mathfrak{B}_t := \|f'(w_t, z_t)\|_*^2 + \|r'(w_t)\|_*^2$, see, e.g., [12]. Then, a non-trivial assumption as $\mathfrak{B}_t \leq G$ for all $t \in \mathbb{N}$ and a $G \in \mathbb{R}$ is imposed to control $\sum_{t=1}^T \eta_t^2\mathfrak{B}_t$ by $O(\sum_{t=1}^T \eta_t^2)$. We refine these discussions by using Assumption 1 to replace $\mathfrak{B}_t$ with $\mathfrak{A}_t$. Equation (3.6) allows us to control $\sum_{t=1}^T \eta_t^2\mathfrak{A}_t$ by $O(\sum_{t=1}^T \eta_t^2)$ without imposing any boundedness assumptions on subgradients. In our discussion for strongly convex objectives, we require to divide both sides of (3.4) by $\eta_t^2$. In this way, Eq. (3.7) plays an analogous role in removing boundedness assumptions in the strongly convex case. Both proofs of Lemma 1 and Lemma 2 are given in Supplementary Material B.

**Lemma 2.** *Let $\{w_t\}_{t\in\mathbb{N}}$ be the sequence produced by (2.2) with $\eta_t \leq (2A)^{-1}\sigma_\Psi$. Then, we have*

$$\|w_{t+1}\|^2 \leq 2C_1\sigma_\Psi^{-1}\sum_{k=1}^t \eta_k, \quad \forall t \in \mathbb{N}, \tag{3.5}$$

*where $C_1 = \sup_{z\in\mathcal{Z}} f(0,z) + r(0) + A^{-1}B$. Furthermore, if $\eta_{t+1} \leq \eta_t$, then for all $t \in \mathbb{N}$*

$$\sum_{k=1}^t \eta_k^2\big(f(w_k,z_k) + r(w_k)\big) \leq 2C_1\sum_{k=1}^t \eta_k^2, \tag{3.6}$$

$$\sum_{k=1}^t \big(f(w_k,z_k) + r(w_k)\big) \leq 2C_1 t + 2C_1\big(\sum_{k=1}^t \eta_k\big)\eta_t^{-1}. \tag{3.7}$$

### 3.1 Convex Objectives

We study the behavior of SCMD for convex objectives with $\sigma_\phi = 0$. The assumption $\sum_{t=1}^\infty \eta_t^2 < \infty$ is satisfied if $\eta_t = \eta_1 t^{-\theta}$ with $\theta > 1/2$ or $\eta_t = \eta_1(t\log^\beta(et))^{-\frac{1}{2}}$ with $\beta > 1$. Our idea is to take a summation of Eq. (3.4) with $w = w^*$, and show that the conditional variance of the involved martingale $\sum_{k=1}^t \eta_k\langle w^* - w_k, f'(w_k,z_k) - \mathbb{E}_{z_k}[f'(w_k,z_k)]\rangle$ can be partially offset by some other terms. The proofs of Theorems 3 and 4 are given in Supplementary Material C.

**Theorem 3.** *Let $\{w_t\}_{t\in\mathbb{N}}$ be the sequence produced by (2.2) with $\eta_t \leq (2A)^{-1}\sigma_\Psi, \eta_{t+1} \leq \eta_t$ and $\sum_{t=1}^\infty \eta_t^2 < \infty$. Then, there exists a constant $C_2$ independent of $T$ (explicitly given in the proof) such that for any $\delta \in (0,1)$ the following inequality holds with probability at least $1 - \delta$*

$$\max_{1\leq t\leq T} \|w_t\|^2 \leq C_2 \log\frac{T}{\delta}. \tag{3.8}$$

**Remark 1.** Although implemented in a possibly unbounded domain, Theorem 3 shows that $\{w_t\}_{t\in\mathbb{N}}$ by (2.2) falls into a bounded ball (up to a logarithmic factor) with high probabilities. Intuitively, this suggests that SCMD is immune to overfitting if we take appropriate step sizes. In this case, we can run SCMD with many iterations without essentially harming the quality of the output model.

Based on Theorem 3, we establish high-probability convergence rates for a weighted average of iterates without any assumptions on the boundedness of iterates. In Theorem 4 and Corollary 5, we establish bounds on suboptimality of objectives w.r.t. any $w$ and an optimal solution $w^*$, respectively.

**Theorem 4.** *Let $w \in \mathcal{W}$ and $\delta \in (0, 2/e)$. Let $\bar{w}_T^{(1)} = \big(\sum_{t=1}^T \eta_t\big)^{-1}\sum_{t=1}^T \eta_t w_t$ be a weighted average of the first $T$ iterates. Under the conditions of Theorem 3, with probability $1 - \delta$ we have*

$$\phi(\bar{w}_T^{(1)}) - \phi(w) \leq \Big(\sum_{t=1}^T \eta_t\Big)^{-1}\big(2C_3 D_\Psi(w,0) + C_4\big)\log^{\frac{3}{2}}\frac{2T}{\delta}, \tag{3.9}$$

*where $C_3$ and $C_4$ are two constants (explicitly given in the proof) independent of $T$.*

**Remark 2.** A similar high-probability bound was established for SCMD in [12]. However, their discussion needs to impose an additional almost-sure boundedness assumption on iterates, i.e., $\|w_t\|_2 \leq G$ for a $G > 0$ and all $t \in \mathbb{N}$. These boundedness assumptions on either subgradients or iterates are fundamental to the existing analysis but hard to check in practice. Moreover, the high-probability analysis makes these assumptions *non-trivial* to remove since one also needs to consider high-order moments of random variables.

**Corollary 5.** *If $\delta \in (0, 2/e)$ and conditions of Theorem 4 are satisfied, then (3.9) holds with probability $1 - \delta$ with $w = w^*$. Furthermore, if we choose $\eta_t = \eta_1 t^{-\theta}$ with $\theta > 1/2$, then with probability $1-\delta$ we have $\phi(\bar{w}_T^{(1)}) - \phi(w^*) = O\big(T^{\theta-1}\log^{\frac{3}{2}}\frac{T}{\delta}\big)$; if we choose $\eta_t = \eta_1(t\log^\beta(et))^{-\frac{1}{2}}$ with $\beta > 1$, then with probability $1-\delta$ we have $\phi(\bar{w}_T^{(1)}) - \phi(w^*) = O\big((T^{-1}\log^\beta T)^{\frac{1}{2}}\log^{\frac{3}{2}}\frac{T}{\delta}\big)$.*

The convergence rate $O\big((T^{-1}\log^\beta T)^{\frac{1}{2}}\log^{\frac{3}{2}}\frac{T}{\delta}\big)$ in Corollary 5 is optimal up to a logarithmic factor [1], which follows directly from Theorem 4 and $\sum_{t=1}^T t^{-\theta} \geq (1-\theta)^{-1}(T^{1-\theta} - 1), \theta \in (0,1)$. We omit the proof for brevity.

In Theorem 6, we give sufficient conditions for the almost sure finiteness of $\lim_{t\to\infty} D_\Psi(w^*, w_t)$ and $\sum_{t=1}^\infty \eta_t\big(\phi(w_t) - \phi(w^*)\big)$. As a direct corollary, we also establish convergence rates with probability one in Corollary 7. Theorem 6 is a part of Proposition E.3 to be presented and proved in Supplementary Material E, while the proof of Corollary 7 is omitted for brevity.

**Theorem 6.** *Consider* $\{w_t\}_{t\in\mathbb{N}}$ *by (2.2) with* $\sum_{t=1}^\infty \eta_t^2 < \infty$. *Then* $\{D_\Psi(w^*, w_t)\}_t$ *converges almost surely (a.s.) to a non-negative random variable and* $\lim_{t\to\infty} D_\Psi(w^*, w_t) < \infty$ *a.s.. Furthermore, if* $\eta_t \leq (2A)^{-1}\sigma_\Psi$ *and* $\eta_{t+1} \leq \eta_t$, *then* $\sum_{t=1}^\infty \eta_t\big(\phi(w_t) - \phi(w^*)\big) < \infty$ *a.s..*

**Corollary 7.** *Let* $\{w_t\}_{t\in\mathbb{N}}$ *be produced by (2.2) and* $\eta_1 \leq (2A)^{-1}\sigma_\Psi$. *If we choose* $\eta_t = \eta_1 t^{-\theta}$ *with* $\theta > 1/2$, *then* $\lim_{T\to\infty} T^{1-\theta}\big(\phi(\bar{w}_T^{(1)}) - \phi(w^*)\big) < \infty$ *a.s.. If we choose* $\eta_t = \eta_1\big(t\log^\beta(et)\big)^{-\frac{1}{2}}$ *with* $\beta > 1$, *then* $\lim_{T\to\infty} \big(\frac{T}{\log^\beta T}\big)^{\frac{1}{2}}\big(\phi(\bar{w}_T^{(1)}) - \phi(w^*)\big) < \infty$ *a.s..*

### 3.2 Strongly Convex Objectives

We now turn to strongly convex objectives with $\sigma_\phi > 0$. In Theorem 8, we establish high-probability bounds for both $\|w_t - w^*\|^2$ and $\phi(\bar{w}_t^{(2)}) - \phi(w^*)$ with $\bar{w}_t^{(2)}$ being another weighted average of the first $t$ iterates, for each of which we derive optimal convergence rates up to a logarithmic factor [1]. The optimality means that not only the dependency on $t$ but also the dependency on the strong-convexity parameter $\sigma_\phi$ can not be improved up to a logarithmic factor [16, 28] ($\sigma_\phi$ is often chosen to be very small in practical learning problems [28, 31]). It should be mentioned that our analysis removes boundedness assumptions on subgradients in the literature [28]. Our idea is to take a weighted summation of (3.4) with $w = w^*$, and show that the conditional variance of an involved martingale $\sum_{k=1}^t (k + t_0 + 1)\langle w^* - w_k, f'(w_k, z_k) - \mathbb{E}_{z_k}[f'(w_k, z_k)]\rangle$ can be partially offset by another term in this weighted summation of (3.4), which is another trick to remove boundedness assumptions on subgradients. We also give a sufficient condition on the almost sure convergence of $w_t$ to $w^*$ in Theorem 9. The proof of Theorem 8 is given in Supplementary Material D. Theorem 9 is a part of Proposition E.3 to be presented in Supplementary Material E.

**Theorem 8.** *Assume* $\sigma_\phi > 0$ *and* $\delta \in (0, e^{-\frac{1}{4}})$. *Let* $\{w_t\}_{t\in\mathbb{N}}$ *be produced by (2.2) with* $\eta_t = \frac{2}{\sigma_\phi t + 2\sigma_F + \sigma_\phi t_0}$, *where* $t_0 \geq \frac{16A\log\frac{T}{\delta}}{\sigma_\phi\sigma_\Psi}$. *Let* $\bar{w}_t^{(2)} = \big(\sum_{k=1}^t(k+t_0+1)\big)^{-1}\sum_{k=1}^t(k+t_0+1)w_k, t \in \mathbb{N}$. *Then, the following inequalities hold with probability* $1 - \delta$ *for all* $t = 1, \ldots, T$

$$\|w^* - w_t\|^2 \leq \frac{C_T}{t + t_0 + 1} \quad and \quad \phi(\bar{w}_t^{(2)}) - \phi(w^*) \leq \frac{\widetilde{C}_T}{t}. \tag{3.10}$$

*Moreover, the dependencies of* $C_T$ *and* $\widetilde{C}_T$ *on* $T/\delta$ *are logarithmic. The dependencies of* $C_T$ *and* $\widetilde{C}_T$ *on* $\sigma_\phi^{-1}$ *are quadratic and linear, respectively.*

**Theorem 9.** *Let* $\{w_t\}_{t\in\mathbb{N}}$ *be the sequence produced by (2.2) with* $\sigma_\phi > 0$. *If* $\sum_{t=1}^\infty \eta_t = \infty$ *and* $\sum_{t=1}^\infty \eta_t^2 < \infty$, *then* $\lim_{t\to\infty} D_\Psi(w^*, w_t) = 0$ *a.s..*

## 4 Generalization Error Bounds

Here we apply our high-probability convergence rates for SCMD to establish generalization error bounds for SGD. In this setting, we assume a training sample $\mathbf{z} = \{z_1, \ldots, z_n\}$ of size $n \in \mathbb{N}$ is drawn independently from a probability measure $\rho$ defined on the sample space $\mathcal{Z}$, and our aim is to learn a hypothesis $h: \mathcal{X} \mapsto \mathbb{R}$ from a hypothesis space $\mathcal{W}$ with good generalization performance. The quality of $h$ at $(x, y)$ is quantified by $\ell(h(x), y)$, where $\ell: \mathbb{R} \times \mathcal{Y} \mapsto \mathbb{R}_+$ is convex w.r.t. the first argument. The generalization error and empirical error of $h$ are defined respectively by $\mathcal{E}(h) = \mathbb{E}_z\big[\ell(h(x), y)\big]$ and $\mathcal{E}_\mathbf{z}(h) = \frac{1}{n}\sum_{i=1}^n \ell\big(h(x_i), y_i\big)$. The best model minimizing the generalization error then becomes $h_\rho = \arg\min_h \mathcal{E}(h)$. We consider a non-parametric learning setting with $\mathcal{W}$ being a reproducing kernel Hilbert space (RKHS) associated to a Mercer kernel $K: \mathcal{X} \times \mathcal{X} \mapsto \mathbb{R}$ which is continuous, symmetric and positive semi-definite [9, 34]. In this learning setting, the candidate models take the form $h_w(x) = \langle w, K_x\rangle$ with $w \in \mathcal{W}$. For brevity, we denote the norm in the RKHS $\mathcal{W}$ by $\|\cdot\|_2$ and introduce abbreviations $\mathcal{E}(w) = \mathcal{E}(h_w), \mathcal{E}_\mathbf{z}(w) = \mathcal{E}_\mathbf{z}(h_w)$. We assume (3.2) and apply the SGD scheme to minimize $\mathcal{E}_\mathbf{z}(w)$. To be specific, we let $w_1 = 0$. At the $t$-th iteration, we randomly choose an index $j_t$ from the uniform distribution over $\{1, \ldots, n\}$ and produce $w_{t+1}$ by

$$w_{t+1} = w_t - \eta_t \ell'\big(\langle w_t, K_{x_{j_t}}\rangle, y_{j_t}\big)K_{x_{j_t}}, \quad t \in \mathbb{N}. \tag{4.1}$$

It is clear that (4.1) is a specific instantiation of (2.2) with $\Psi(w) = \frac{1}{2}\|w\|_2^2$, $f(w,z) = \ell(\langle w, K_x \rangle, y)$, $r(w) = 0$ and $\tilde{\rho}$ in Section 2 being the uniform distribution over $\{z_1, \ldots, z_n\}^2$. Therefore, the objective function to which SGD is applied becomes $\phi(w) = \mathcal{E}_{\mathbf{z}}(w)$.

To state our generalization bounds, we need to introduce an assumption on a polynomial decay rate of approximation errors.

**Assumption 3.** We assume the approximation error $D(\lambda) := \inf_{w \in \mathcal{W}} \mathcal{E}(w) - \mathcal{E}(h_\rho) + \lambda\|w\|_2^2$ enjoys a polynomial decay with exponent $0 < \alpha \leq 1$ in the sense $D(\lambda) \leq c_\alpha \lambda^\alpha, \forall \lambda > 0$, where $c_\alpha > 0$.

**Remark 3.** Assumption 3 is standard in learning theory and satisfied under some mild conditions on the smoothness of the function $h_\rho$ and the representation power of $\mathcal{W}$ [9, 33]. If $\ell$ is smooth, then $D(\lambda)$ can be controlled by $\widetilde{D}(\lambda) := \inf_{w \in \mathcal{W}} \|h_w - h_\rho\|_{L^2_{\rho_\mathcal{X}}}^2 + \lambda\|w\|_2^2$, which quantifies the approximation of $h_\rho$ by RKHS in $L^2_{\rho_\mathcal{X}}$ (square-integrable function class with marginal measure $\rho_\mathcal{X}$) and is well studied in approximation theory. $\widetilde{D}(\lambda)$ decays polynomially with $\alpha \in (0,1]$ if $h_\rho \in L_K^{\alpha/2}(L^2_{\rho_\mathcal{X}})$, where $L_K : L^2_{\rho_\mathcal{X}} \mapsto L^2_{\rho_\mathcal{X}}$ is the integral operator associated to $K$ [9, Proposition 8.5]. Similar results hold if $\ell$ is Lipschitz continuous. Assumption 3 also holds if we use Gaussian kernels with flexible variances and distributions with geometric noise conditions [35]. It should be mentioned that kernels need not to be universal for Assumption 3 since it concerns the target function $h_\rho$, which may admit more regularity (e.g., expressed by $L_K$) than continuity, while universality means that $D(\lambda) \to 0$ as $\lambda \to 0$ for *all continuous* $h_\rho$ [34].

We now establish a generalization error bound for a weighted average of iterates produced by (4.1) to be proved in Supplementary Material F, which is derived by decomposing the excess generalization error $\mathcal{E}(\bar{w}_T^{(1)}) - \mathcal{E}(h_\rho)$ into three components: an estimation error, an approximation error and a computational error. As we will see in the proof, the term $\left(\sum_{t=1}^T \eta_t\right)^{-\alpha}$ is due to the approximation and computational error, while the term $n^{-\frac{\alpha}{1+\alpha}}$ is due to the estimation and approximation error. The bound becomes $n^{-\frac{\alpha}{1+\alpha}} \log^{\frac{3}{2}} \frac{8T}{\delta}$ for sufficiently large $T$, which enjoys a logarithmic dependency on $T$ and demonstrates the ability of SGD to avoid overfitting.

**Theorem 10.** *Let $\{w_t\}_{t \in \mathbb{N}}$ be the sequence produced by (4.1) with $\eta_t \leq (2A)^{-1}\sigma_\Psi, \eta_{t+1} \leq \eta_t$ and $\sum_{t=1}^\infty \eta_t^2 < \infty$. Suppose Assumption 3 holds. Then, for any $T$ satisfying $\sum_{t=1}^T \eta_t \geq 1$ and $\delta \in (0, 2/e)$, the following inequality holds with probability at least $1 - \delta$*

$$\mathcal{E}(\bar{w}_T^{(1)}) - \mathcal{E}(h_\rho) \leq C_5 \max\left\{ \left(\sum_{t=1}^T \eta_t\right)^{-\alpha}, n^{-\frac{\alpha}{1+\alpha}} \right\} \log^{\frac{3}{2}} \frac{8T}{\delta}, \tag{4.2}$$

*where $C_5$ is a constant independent of $T$ (explicitly given in the proof).*

We consider specific step sizes in Theorem 10 and choose an appropriate time index to get concrete generalization bounds, as shown in Corollary 11. The bound $O\big(n^{-\frac{\alpha}{1+\alpha}} \log^{\frac{3+\alpha\beta}{2}} \frac{n}{\delta}\big)$ coincides with $O(n^{-\frac{\alpha}{1+\alpha}} \log n)$ (up to a logarithmic factor) in expectation for convex and smooth loss functions [21], and largely improves the bound $O(n^{-\frac{\alpha}{1+2\alpha}} \log n)$ in expectation for convex and non-smooth loss functions [21]. In particular, if $\alpha = 1$ we derive the optimal bound $O(n^{-\frac{1}{2}} \log^{\frac{3+\beta}{2}} \frac{n}{\delta})$ in a general case with neither Bernstein conditions on variances nor capacity assumptions on hypothesis spaces (up to a logarithmic factor). It is also clear that SGD with different step sizes can achieve similar generalization bounds. However, the computational complexity to fulfill this statistical potential can be significantly different. Corollary 11, with the proof omitted, follows directly from Theorem 10 and $\sum_{t=1}^T t^{-\theta} \geq (1-\theta)^{-1}(T^{1-\theta} - 1), \theta \in (0,1)$. Denote $\lceil a \rceil$ the least integer no less than $a$.

**Corollary 11.** *Consider $\{w_t\}_{t \in \mathbb{N}}$ by (4.1) and $\delta \in (0, 2/e)$. Let Assumption 3 hold and $\sum_{t=1}^T \eta_t \geq 1$.*

*(a) If we take $\eta_t = \eta_1 t^{-\theta}$ with $\eta_1 \leq (2A)^{-1}$ and $\theta \in (1/2, 1)$, then with probability $1 - \delta$ that*

$$\mathcal{E}(\bar{w}_T^{(1)}) - \mathcal{E}(h_\rho) = O\left( \left( T^{-\alpha(1-\theta)} + n^{-\frac{\alpha}{1+\alpha}} \right) \log^{\frac{3}{2}} \frac{T}{\delta} \right).$$

*If we further take $T^* = \left\lceil n^{\frac{1}{(1+\alpha)(1-\theta)}} \right\rceil$, then we get $\mathcal{E}(\bar{w}_{T^*}^{(1)}) - \mathcal{E}(h_\rho) = O\big(n^{-\frac{\alpha}{1+\alpha}} \log^{\frac{3}{2}} \frac{n}{\delta}\big)$.*

*(b) If we take $\eta_t = \eta_1(t \log^\beta(et))^{-\frac{1}{2}}$ with $\eta_1 \leq (2A)^{-1}$ and $\beta > 1$, then with probability $1 - \delta$ that*

$$\mathcal{E}(\bar{w}_T^{(1)}) - \mathcal{E}(h_\rho) = O\bigg( \Big( T^{-\frac{\alpha}{2}} \log^{\frac{\alpha\beta}{2}} T + n^{-\frac{\alpha}{1+\alpha}} \Big) \log^{\frac{3}{2}} \frac{T}{\delta} \bigg).$$

*If we further take $T^* = \big\lceil n^{\frac{2}{1+\alpha}} \big\rceil$, then we get $\mathcal{E}(\bar{w}_{T^*}^{(1)}) - \mathcal{E}(h_\rho) = O\big( n^{-\frac{\alpha}{1+\alpha}} \log^{\frac{3+\alpha\beta}{2}} \frac{n}{\delta} \big).$*

It should be noted that our discussions depend on the existence of a minimizer of $\mathcal{E}_{\mathbf{z}}(\cdot)$ over the RKHS with a finite norm. This assumption can be relaxed to the existence of a minimizer of $\mathcal{E}(\cdot)$ over the RKHS with a finite norm to derive similar generalization bounds. Indeed, one can perform deductions similar to the proof of Theorem 3 by taking $w$ in (3.4) to be the minimizer of $\mathcal{E}(\cdot)$. However, in this case it becomes a challenge to derive estimation error bounds with a logarithmic dependency on $T$.

# 5 Related Work and Discussions

## 5.1 Convex Objectives

For general convex objectives, regret bounds $O(\sqrt{T})$ were established for online gradient descent with $T$ iterations [44], from which one can directly derive convergence rates $O(T^{-\frac{1}{2}})$ for SGD with some averaging schemes. This result was extended to stochastic forward-backward splitting [11]. A convergence rate $O(T^{-\frac{1}{2}} \log T)$ was established for the $T$-th individual iterate of SGD [32]. All the above mentioned rates were stated in expectation and derived based on an assumption $\mathbb{E}[\|f'(w_t, z_t)\|_*^2 + \|r'(w_t)\|_*^2] \leq G$ for a $G \geq 0$ and $t \in \mathbb{N}$. This boundedness assumption was successfully removed for studying convergence rates in expectation under some smoothness assumption [23, 40, 42] or Assumption 1 [30]. As compared to these convergence rates in expectation, high-probability convergence rates were much less studied and were often based on a stronger assumption on the almost sure boundedness of subgradients. Under the assumption $\max\{D_\Psi(w^*, w_t), \sup_z \|f'(w_t, z)\|_*\} \leq G$ for a $G > 0$ and all $t \in \mathbb{N}$, it was shown with probability $1 - \delta$ that $\phi(\bar{w}_T^{(1)}) - \phi(w^*) = O\big(T^{-\frac{1}{2}} \log^{\frac{1}{2}} \frac{1}{\delta}\big)$ for $\bar{w}_T^{(1)}$ defined in Theorem 4 [12, 24]. High-probability bounds were also established for stochastic dual averaging under the boundedness assumption on iterates and subgradients [37]. In our discussion, we show that the same high-probability convergence rate (up to a logarithmic factor) holds without any boundedness assumptions on either the iterates $\{w_t\}$ or the associated subgradients. In particular, we show that $\{w_t\}_{t \leq T}$ automatically falls into a ball with radius $O(\sqrt{\log T/\delta})$ with high probability. It was shown with probability $1 - \delta$ that $\|w_t - w^*\|_2^2 = O(\|w^*\|_2^2 \log \frac{T}{\delta})$ for the particular SGD [19]. However, the discussion in [19] requires a stronger assumption on the Hölder continuity of loss functions which excludes non-differentiable loss functions such as hinge loss and the absolute loss satisfying (3.2). Secondly, they only consider the one-pass SGD where each training example is used only once.

We also give a sufficient condition for almost sure finiteness of $\sum_{t=1}^\infty \eta_t\big(\phi(w_t) - \phi(w^*)\big)$, while most results on almost sure convergence are achieved for strongly convex objectives.

## 5.2 Strongly Convex Objectives

For $\lambda$-exp-concave loss functions, a regret bound $O(\lambda^{-1} \log T)$ was established for an online Newton method [15], which implies convergence rates $O\big((\lambda T)^{-1} \log T\big)$ for some average of iterates produced by the stochastic counterpart. This result was extended to online forward-backward splitting [11] and SCMD [12] applied to $\lambda$-strongly convex objectives. Optimal convergence rates $O((\lambda T)^{-1})$ for the suboptimality of objective values were derived based on a suffix averaging scheme [28], a epoch-GD scheme based on a doubling trick [14] and a weighted averaging with a weight of $t + 1$ for $w_t$ [16]. However, the above mentioned results are all associated to convergence rates in expectation and require to impose boundedness assumptions on subgradients encountered during the iterations. This boundedness assumption was relaxed as $\mathbb{E}_z[\|f'(w_t, z)\|_*^2] \leq A_1 + B_1 \|F'(w_t)\|_*^2$ for SGD [6] with $A_1, B_1 \geq 0$, which was further removed for SGD [26] and stochastic mirror descent [17] by imposing smoothness assumptions on loss functions. All the above mentioned results are stated in expectation. With probability $1 - \delta$, it was shown $\|w_T - w^*\|^2 = O\big((\lambda^2 T)^{-1} + (\lambda T)^{-1} \log(\delta^{-1} \log T)\big)$ for SGD [28]. High-probability convergence rates $O\big((\lambda T)^{-1} \log(\delta^{-1} \log T)\big)$ were also established for the suboptimality of objective values for the $T$-th iterate of the epoch-GD [14]. These two high-probability rates were derived based on an assumption on almost sure boundedness of subgradients

which is more challenging to remove [14, 28]. As a comparison, we establish the same convergence rate (up to a logarithmic factor) for a more general SCMD without boundedness assumptions on subgradients. Sufficient conditions as in Theorem 9 were established for almost sure convergence of SGD [5, 26] and stochastic mirror descent [17], which were extended to SCMD in Theorem 9.

## 5.3 Generalization Error Bounds

While computational complexity of SGD has been extensively studied in the optimization community, there is much less work on the generalization property of the model trained by SGD. Classical generalization bounds only hold for one-pass SGD [24, 27, 28, 32, 36, 38, 39] where each training example can be used at most once. In practice, however, multiple passes are often used to produce a model with good generalization behavior [13]. The landmark work in [7] developed a framework to analyze generalization performance of multi-pass stochastic learning algorithms by taking into account the computational complexity of learning algorithms. Under this framework, the interplay among estimation errors, computational errors and approximation errors can be studied, showing that an implicit regularization can be achieved in the absence of penalization or constraints by tuning either the step size or the number of passes (the iteration number divided by the training set size) [13, 20, 21, 29]. In a parametric setting, it was shown that SGD is algorithmically stable and the stability measure of SGD with $T$ iterates scales as $O(n^{-1} \sum_{t=1}^{T} \eta_t)$ [13], based on which a generalization bound $\mathbb{E}[\mathcal{E}(\bar{w}_T^{(1)})] - \inf_{w \in \mathcal{W}} \mathcal{E}(w) = O(n^{-\frac{1}{2}})$ was established for $\eta_t = O(1/\sqrt{n})$ and $T = O(n)$ without considering approximation errors. The discussion in [13] requires to impose a smoothness assumption on loss functions. Generalization analysis was considered separately for smooth and non-smooth loss functions [21]. For smooth loss functions, it was shown $\mathbb{E}[\mathcal{E}(\bar{w}_T^{(1)})] - \mathcal{E}(h_\rho) = O(n^{-\frac{\alpha}{1+\alpha}} \log n)$ for $\eta_t = \eta_1/\sqrt{t}$ with $T = \lceil n^{\frac{2}{\alpha+1}} \rceil$ [21], based on the stability property of SGD established in [13]. For non-smooth loss functions, it was shown $\mathbb{E}[\mathcal{E}(\bar{w}_T^{(1)})] - \mathcal{E}(h_\rho) = O(n^{-\frac{\alpha}{2\alpha+1}} \log n)$ for $\eta_t = \eta_1/\sqrt{t}$ and $T = \lceil n^{\frac{2}{2\alpha+1}} \rceil$, by controlling estimation errors with Rademacher complexities [3, 21]. Still, the bounds in [13, 21] require to impose a boundedness assumption on subgradients and are stated in expectation. As a comparison, we establish high-probability bounds without any boundedness assumptions on subgradients. Furthermore, our generalization analysis extends the analysis in [13] to non-smooth loss functions and substantially improve the bound $O(n^{-\frac{\alpha}{2\alpha+1}} \log n)$ [21] in this setting. The generalization error bound $O\big(n^{-\frac{\alpha}{1+\alpha}} \log^{\frac{3+\alpha\beta}{2}} \frac{n}{\delta}\big)$ in Corollary 11 is optimal in the sense that it matches the best available bound for Tikhonov regularization (up to a logarithmic factor) [9, 21, 34].

We achieve this improvement by controlling better estimation errors. Specifically, estimation errors were shown to scale polynomially w.r.t. the number of passes [13, 21], which dominate the other two errors for large $T$. In this way, one needs to tune $T$ to balance the estimation, approximation and computational errors. As a comparison, we show bounds scaling logarithmically w.r.t. the number of passes for $\mathcal{E}(\bar{w}_T^{(1)}) - \mathcal{E}_{\mathbf{z}}(\bar{w}_T^{(1)})$ (Theorem 10). This implies that estimation errors will never essentially dominate the other two errors and one can run SGD with a sufficient number of passes with little overfitting if step sizes are square-summable, due to the key observation on the almost boundedness of iterates established in Theorem 3. Another trick in getting almost optimal bounds includes the use of Assumption 3 to control $\mathcal{E}(w_\lambda) - \mathcal{E}_{\mathbf{z}}(w_\lambda)$ with a linear (instead of quadratic) function of $\sup_z f(w_\lambda, z)$ and to select a suitable $\lambda$, where $w_\lambda = \arg\min_{w \in \mathcal{W}} \mathcal{E}(w) + \lambda \|w\|_2^2$. Optimal learning rates were given for multi-pass SGD with the least squares loss function [10, 20, 29]. However, their analysis is based on an integral operator approach and does not apply to general loss functions. Generalization bounds for SGD were also studied from a PAC-Bayesian perspective [22]. However, the high-probability bounds there require to impose Lipschitz continuity, smoothness and strong convexity assumptions on loss functions, and ignore computational and approximation errors [22].

## 6 Simulations

Our analysis implies that SGD can be run with a sufficient number of iterations with little overfitting if step sizes are square-summable, which meanwhile can achieve similar generalization performance with different computational complexities. In this section, we include some experimental results to validate these theoretical findings. We apply SGD (4.1) with a linear kernel $K_x = x$ and the hinge loss $\ell(a, y) = \max\{0, 1 - ya\}$ to several binary classification datasets (ADULT, GISETTE, IJCNN, MUSHROOMS, PHISHING and SPLICE). All these datasets, described in Supplementary Material

G, can be download from the LIBSVM website [8]. We consider polynomially decaying step sizes of the form $\eta_t = 5t^{-\theta}$ with $\theta \in \{0.25, 0.51, 0.75\}$ (we consider $\theta = 0.51$, instead of $\theta = 0.5$, since the associated step size sequence is square-summable). We repeat experiments 12 times and report the average of results. In Figure 1, we plot test errors of $\bar{w}_t^{(3)} = \left( \sum_{k=\tilde{t}+1}^{t} \eta_k \right)^{-1} \sum_{k=\tilde{t}+1}^{t} \eta_k w_k$ versus the number of passes (the iteration number divided by the training set size), where $\tilde{t} = 2^{\lfloor \log_2 t \rfloor - 1}$. Intuitively, $\bar{w}_t^{(3)}$ returns an $\alpha$-suffix average of iterates [28] with $\alpha \in [1/2, 3/4]$ and one can adapt the proof of Theorem 4 to show that $\bar{w}_t^{(3)}$ enjoys similar generalization bounds as $\bar{w}_t^{(1)}$. Moreover, $\bar{w}_t^{(3)}$ is easily computable on-the-fly by storing only $\sum_{j=1}^{k} \eta_j w_j$ with $k = 2^0, 2^1, 2^2, \ldots$. From Figure 1, we see that SGD is resistant to overfitting for appropriate step sizes. For example, we observe no overfitting even if the number of passes exceeds 1000 for SGD with $\theta \in \{0.51, 0.75\}$. Moreover, SGD with $\theta \in \{0.51, 0.75\}$ can achieve similar generalization errors on ADULT, IJCNN, PHISHING and SPLICE, towards which SGD with $\theta = 0.51$ requires a significantly smaller number of passes. This is well consistent with Corollary 11.

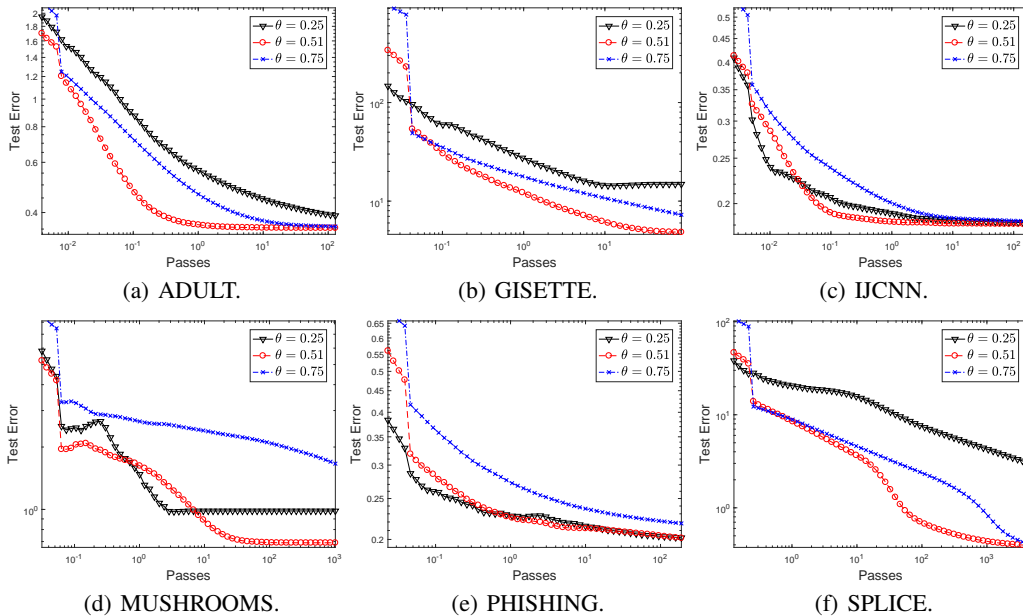

Figure 1: Test errors versus the number of passes.

## 7 Conclusions

In this paper, we establish a rigorous theoretical foundation for SCMD by providing optimal convergence rates (up to a logarithmic factor) in the stochastic optimization setting without boundedness assumptions on either subgradients or iterates, which in turn also shed new insights on the generalization behavior of the multi-pass SGD in the statistical learning theory setting. In particular, we justify the immunity of multi-pass SGD to overfitting by giving estimation error bounds with a *logarithmic* dependency on the number of passes for square-summable step sizes, while existing bounds scale *polynomially* [13, 21]. This improvement is based on the key observation on the almost boundedness of iterates with high probability. Our generalization analysis of SGD also substantially improves learning rates in [21], removes bounded subgradient assumptions in [13, 21, 22], removes smoothness assumptions in [13, 22] and is performed in high probability instead of in expectation [13, 21]. It would be interesting to extend our results to a non-convex setting [43] and to general mirror descent algorithms with a *non-differentiable* mirror map [18].

**Acknowledgments**

This work is supported in part by the National Key Research and Development Program of China (Grant No. 2017YFB1003102), the National Natural Science Foundation of China (Grant

Nos. 61806091 and 61672478), the Science and Technology Innovation Committee Foundation of Shenzhen (Grant No. ZDSYS201703031748284) and Shenzhen Peacock Plan (Grant No. KQTD2016112514355531).

## Footnotes

[2]$\rho$ is related to the draw of training examples while $\tilde{\rho}$ is related to the draw of indices for SGD.

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
