[Supplementary Material]

# Supplementary Material

**Yunwen Lei** and **Ke Tang**[*]
Shenzhen Key Laboratory of Computational Intelligence, Department of Computer Science
and Engineering, Southern University of Science and Technology, Shenzhen 518055, China
leiyw@sustc.edu.cn tangk3@sustc.edu.cn

## A    Technical Lemmas

### A.1    Concentration Inequalities

Our discussion on high-probability bounds is based on the following two concentration inequalities. Lemma A.1 quantifies the concentration behavior of martingales. Part (a) is the Azuma-Hoeffding inequality for martingales with bounded increments [4], and part (b) is a conditional Bernstein inequality using the conditional variance to quantify better the concentration behavior of martingales [10]. Lemma A.2 is the McDiarmid's inequality to arbitrary real-valued functions of independent random variables that satisfy a bounded increment condition [6].

**Lemma A.1.** *Let $z_1, \ldots, z_n$ be a sequence of random variables such that $z_k$ may depend on the previous random variables $z_1, \ldots, z_{k-1}$ for all $k = 1, \ldots, n$. Consider a sequence of functionals $\xi_k(z_1, \ldots, z_k), k = 1, \ldots, n$. Let $\sigma_n^2 = \sum_{k=1}^n \mathbb{E}_{z_k}\left[\left(\xi_k - \mathbb{E}_{z_k}[\xi_k]\right)^2\right]$ be the conditional variance.*

(a) *Assume that $|\xi_k - \mathbb{E}_{z_k}[\xi_k]| \leq b_k$ for each $k$. Let $\delta \in (0, 1)$. With probability at least $1 - \delta$ we have*

$$\sum_{k=1}^n \xi_k - \sum_{k=1}^n \mathbb{E}_{z_k}[\xi_k] \leq \left(2 \sum_{k=1}^n b_k^2 \log \frac{1}{\delta}\right)^{\frac{1}{2}}. \tag{A.1}$$

(b) *Assume that $\xi_k - \mathbb{E}_{z_k}[\xi_k] \leq b$ for each $k$. Let $\rho \in (0, 1]$ and $\delta \in (0, 1)$. With probability at least $1 - \delta$ we have*

$$\sum_{k=1}^n \xi_k - \sum_{k=1}^n \mathbb{E}_{z_k}[\xi_k] \leq \frac{\rho \sigma_n^2}{b} + \frac{b \log \frac{1}{\delta}}{\rho}. \tag{A.2}$$

**Lemma A.2.** *Let $c_1, \ldots, c_n \in \mathbb{R}_+$. Let $Z_1, \ldots, Z_n$ be independent random variables taking values in a set $\mathcal{Z}$, and assume that $f : \mathcal{Z}^n \to \mathbb{R}$ satisfies*

$$\sup_{z_1, \ldots, z_n, \bar{z}_k \in \mathcal{Z}} |f(z_1, \cdots, z_n) - f(z_1, \cdots, z_{k-1}, \bar{z}_k, z_{k+1}, \cdots, z_n)| \leq c_k \tag{A.3}$$

*for $k = 1, \ldots, n$. Then, for any $0 < \delta < 1$, with probability at least $1 - \delta$ we have*

$$f(Z_1, \ldots, Z_n) \leq \mathbb{E}\big[f(Z_1, \ldots, Z_n)\big] + \sqrt{\frac{\sum_{k=1}^n c_k^2 \log(1/\delta)}{2}}.$$

### A.2    Behavior of Objectives

In this section, we collect some lemmas on functions $g$ satisfying

$$\|g'(w)\|_*^2 \leq A g(w) + B \tag{A.4}$$

for some constant $A, B \geq 0$. Lemma A.3 shows that, if $g$ satisfies (A.4), then both $\|g'(w)\|_*^2$ and $g(w)$ can be controlled by quadratic functions of $\|w\|$.

---

[*]Corresponding author

**Lemma A.3.** *Let $g : \mathcal{W} \mapsto \mathbb{R}$ be a convex function. If there exist $A$ and $B$ such that* (A.4) *holds for all $w \in \mathcal{W}$. Then*

$$\|g'(w)\|_*^2 \leq 2A^2\|w\|^2 + 2Ag(0) + 2B \quad and \quad g(w) \leq \left(A^2 + \frac{1}{2}\right)\|w\|^2 + (A+1)g(0) + B. \quad \text{(A.5)}$$

*Proof.* According to (A.4) and the convexity of $g$, we know
$$\|g'(w)\|_*^2 \leq A\big(g(w) - g(0)\big) + Ag(0) + B$$
$$\leq A\langle w, g'(w)\rangle + Ag(0) + B \leq A\|w\|\|g'(w)\|_* + Ag(0) + B.$$
Solving the above quadratic inequality of $\|g'(w)\|_*$ shows
$$\|g'(w)\|_* \leq A\|w\| + \sqrt{Ag(0) + B},$$
from which and the elementary inequality $(a+b)^2 \leq 2(a^2 + b^2)$ we derive the first inequality.

We now turn to the second inequality. By the convexity of $g$ and the first inequality in (A.5), we get
$$g(w) - g(0) \leq \langle w, g'(w)\rangle \leq \|w\|\|g'(w)\|_*$$
$$\leq \frac{\|w\|^2}{2} + \frac{\|g'(w)\|_*^2}{2} \leq \frac{\|w\|^2}{2} + A^2\|w\|^2 + Ag(0) + B,$$
from which we derive the second inequality. The proof is complete. $\qquad\square$

Lemma A.4 shows that functions of the form $f(w, z) = \ell(\langle w, x\rangle, y)$ would satisfy (3.1) if $\ell$ satisfies (A.6).

**Lemma A.4.** *Let $\ell : \mathbb{R} \times \mathcal{Y} \mapsto \mathbb{R}$ and $f(w, z) = \ell(\langle w, x\rangle, y)$ with $z = (x, y)$. If there exist $\tilde{A}, \tilde{B} \geq 0$ such that*
$$|\ell'(a, y)|^2 \leq \tilde{A}\ell(a, y) + \tilde{B}, \quad \forall a \in \mathbb{R}, y \in \mathcal{Y}. \quad \text{(A.6)}$$
*Then we have $\|f'(w, z)\|_*^2 \leq Af(w, z) + B$ for any $w \in \mathcal{W}$ and $z \in \mathcal{Z}$, where $\kappa = \sup_{x \in \mathcal{X}} \|x\|_*$, $A = \tilde{A}\kappa^2$ and $B = \tilde{B}\kappa^2$.*

*Proof.* For any $w \in \mathcal{W}$ and $z \in \mathcal{Z}$, it follows from (A.6) that
$$\|f'(w, z)\|_*^2 = \big\|\ell'(\langle w, x\rangle, y)x\big\|_*^2 \leq \kappa^2\Big(\tilde{A}\ell(\langle w, x\rangle, y) + \tilde{B}\Big) = \kappa^2\big(\tilde{A}f(w, z) + \tilde{B}\big).$$
The proof is complete. $\qquad\square$

Lemma A.5 shows that regularizers $r_p(w) = \|w\|_p^p, p \in [1, 2]$ satisfy the condition (3.1). For $a \in \mathbb{R}$, denote by $\mathrm{sgn}(a)$ the sign of $a$, i.e., $\mathrm{sgn}(a) = 1$ if $a > 0$, $\mathrm{sgn}(a) = -1$ if $a < 0$ and $\mathrm{sgn}(a) = 0$ if $a = 0$.

**Lemma A.5.** *The function $r_p(w) = \|w\|_p^p$ with $1 \leq p \leq 2$ defined on $\mathcal{W}$ satisfies*
$$\|r_p'(w)\|_{p^*}^2 \leq p\big(2(p-1)\|w\|_p^p + 2 - p\big), \quad \forall w \in \mathcal{W},$$
*where $p^* = \frac{p}{p-1}$ is the conjugate exponent of $p$.*

*Proof.* If $p = 1$, then any $r_1'(w) \in \partial r_1(w)$ would satisfy $\|r_1'(w)\|_\infty \leq 1$, from which and $p^* = \infty$ we know $\|r_1'(w)\|_{p^*}^2 \leq 1$.

If $p > 1$, then the gradient of $r_p$ at $w$ can be calculated by $\nabla r_p(w) = p\big(\mathrm{sgn}(w(i))|w(i)|^{p-1}\big)_{i=1}^d$, from which we have

$$\|\nabla r_p(w)\|_{p^*} = p\Big(\sum_{i=1}^d \big|\mathrm{sgn}(w(i))|w(i)|^{p-1}\big|^{p^*}\Big)^{\frac{1}{p^*}} = p\Big(\sum_{i=1}^d |w(i)|^{p^*(p-1)}\Big)^{\frac{1}{p^*}} = p\|w\|_p^{p-1}.$$

It then follows from the Young's inequality
$$ab \leq \frac{a^s}{s} + \frac{b^{\tilde{s}}}{\tilde{s}}, \quad \forall a, b, s, \tilde{s} > 0 \text{ with } \frac{1}{s} + \frac{1}{\tilde{s}} = 1$$

that

$$\|\nabla r_p(w)\|_{p^*}^2 = p^2\|w\|_p^{2(p-1)} \leq p^2\left(\frac{\|w\|_p^{2(p-1)\frac{p}{2(p-1)}}}{\frac{p}{2(p-1)}} + \frac{2-p}{p}\right) = p\big(2(p-1)\|w\|_p^p + 2 - p\big).$$

The proof is complete by combining the above two cases together. $\qquad\square$

# B   Proofs for Lemma 1 and Lemma 2

In this section, we prove Lemma 1 quantifying the one-step progress of SCMD (2.2), and Lemma 2 which plays an important role in removing the boundedness assumptions on subgradients.

*Proof of Lemma 1.* According to the first-order optimality condition in (2.2), there exists an $r'(w_{t+1}) \in \partial r(w_{t+1})$ satisfying

$$\eta_t f'(w_t, z_t) + \eta_t r'(w_{t+1}) + \nabla\Psi(w_{t+1}) - \nabla\Psi(w_t) = 0,$$

from which and the identity $D_\Psi(w, w_{t+1}) + D_\Psi(w_{t+1}, w_t) - D_\Psi(w, w_t) = \langle w - w_{t+1}, \nabla\Psi(w_t) - \nabla\Psi(w_{t+1})\rangle$, we derive

$$
\begin{aligned}
D_\Psi(w, w_{t+1}) - D_\Psi(w, w_t) &= D_\Psi(w, w_{t+1}) + D_\Psi(w_{t+1}, w_t) - D_\Psi(w, w_t) - D_\Psi(w_{t+1}, w_t) \\
&= \langle w - w_{t+1}, \nabla\Psi(w_t) - \nabla\Psi(w_{t+1})\rangle - D_\Psi(w_{t+1}, w_t) \\
&= \eta_t \langle w - w_{t+1}, f'(w_t, z_t) + r'(w_{t+1})\rangle - D_\Psi(w_{t+1}, w_t) \\
&\leq \eta_t \langle w - w_{t+1}, f'(w_t, z_t)\rangle + \eta_t\big[r(w) - r(w_{t+1}) - \sigma_r D_\Psi(w, w_{t+1})\big] - D_\Psi(w_{t+1}, w_t) \\
&= \eta_t \langle w - w_t, f'(w_t, z_t)\rangle + \eta_t \langle w_t - w_{t+1}, f'(w_t, z_t)\rangle + \eta_t[r(w) - r(w_t)] \\
&\quad + \eta_t[r(w_t) - r(w_{t+1})] - \sigma_r \eta_t D_\Psi(w, w_{t+1}) - D_\Psi(w_{t+1}, w_t).
\end{aligned}
$$

$$\text{(B.1)}$$

Here, we have used the $\sigma_r$-strong convexity of $r$ (3.3) in the inequality. From the convexity of $r$, the definition of dual norm and the strong convexity of $\Psi$, it follows that

$$
\begin{aligned}
&\eta_t\big[\langle w_t - w_{t+1}, f'(w_t, z_t)\rangle + r(w_t) - r(w_{t+1})\big] - D_\Psi(w_{t+1}, w_t) \\
&\leq \eta_t\|w_t - w_{t+1}\|\|f'(w_t, z_t)\|_* + \eta_t\langle w_t - w_{t+1}, r'(w_t)\rangle - 2^{-1}\sigma_\Psi\|w_t - w_{t+1}\|^2 \\
&\leq \eta_t\|w_t - w_{t+1}\|\big[\|f'(w_t, z_t)\|_* + \|r'(w_t)\|_*\big] - 2^{-1}\sigma_\Psi\|w_t - w_{t+1}\|^2 \\
&\leq 2^{-1}\sigma_\Psi\|w_t - w_{t+1}\|^2 + 2^{-1}\sigma_\Psi^{-1}\eta_t^2\big[\|f'(w_t, z_t)\|_* + \|r'(w_t)\|_*\big]^2 - 2^{-1}\sigma_\Psi\|w_t - w_{t+1}\|^2 \\
&\leq \sigma_\Psi^{-1}\eta_t^2\big[\|f'(w_t, z_t)\|_*^2 + \|r'(w_t)\|_*^2\big] \leq \sigma_\Psi^{-1}\eta_t^2\big[Af(w_t, z_t) + Ar(w_t) + 2B\big],
\end{aligned}
$$

where we have used the elementary inequality $(a + b)^2 \leq 2(a^2 + b^2)$ and (3.1) in the last two inequalities. Plugging the above inequality back into (B.1), we get the stated inequality and complete the proof. □

*Proof of Lemma 2.* Using the convexity of $f$ in (3.4), we derive the following inequality for any $w \in \mathcal{W}$

$$
\begin{aligned}
&D_\Psi(w, w_{t+1}) - D_\Psi(w, w_t) \\
&\leq \eta_t\big(f(w, z_t) - f(w_t, z_t)\big) + \eta_t(r(w) - r(w_t)) + \sigma_\Psi^{-1}\eta_t^2\big(Af(w_t, z_t) + Ar(w_t) + 2B\big) \\
&= \eta_t(f(w, z_t) + r(w)) + (\sigma_\Psi^{-1}\eta_t^2 A - \eta_t)\big(f(w_t, z_t) + r(w_t)\big) + 2\sigma_\Psi^{-1}B\eta_t^2 \\
&\leq \eta_t(f(w, z_t) + r(w)) + A^{-1}B\eta_t,
\end{aligned}
$$

$$\text{(B.2)}$$

where the last inequality is due to the assumption $\eta_t \leq (2A)^{-1}\sigma_\Psi$. Plugging $w = 0$ in the above inequality and using the definition of $C_1$, we derive

$$D_\Psi(0, w_{t+1}) - D_\Psi(0, w_t) \leq \eta_t(f(0, z_t) + r(0)) + A^{-1}B\eta_t \leq \eta_t C_1.$$

It then follows that

$$D_\Psi(0, w_{t+1}) = D_\Psi(0, w_1) + \sum_{k=1}^{t}\big[D_\Psi(0, w_{k+1}) - D_\Psi(0, w_k)\big] \leq C_1 \sum_{k=1}^{t}\eta_k, \qquad \text{(B.3)}$$

where we have used $w_1 = 0$ in the last inequality. The stated inequality (3.5) then follows from the $\sigma_\Psi$-strong convexity of $\Psi$.

We now prove (3.6). Taking $w = 0$ in (B.2) and using $\eta_t \leq 2^{-1}A^{-1}\sigma_\Psi$, we get

$$2^{-1}\eta_t\big(f(w_t, z_t) + r(w_t)\big) \leq \eta_t\big(f(0, z_t) + r(0)\big) + 2\sigma_\Psi^{-1}B\eta_t^2 + D_\Psi(0, w_t) - D_\Psi(0, w_{t+1}). \quad \text{(B.4)}$$

Multiplying both sides by $2\eta_t$ then gives

$$
\begin{aligned}
\eta_t^2\big(f(w_t,z_t)+r(w_t)\big) &\leq 2\eta_t^2\big(f(0,z_t)+r(0)\big)+4\sigma_\Psi^{-1}B\eta_t^3+2\eta_t\big(D_\Psi(0,w_t)-D_\Psi(0,w_{t+1})\big)\\
&\leq 2\eta_t^2\big(f(0,z_t)+r(0)\big)+2A^{-1}B\eta_t^2+2\eta_t D_\Psi(0,w_t)-2\eta_{t+1}D_\Psi(0,w_{t+1})\\
&\leq 2C_1\eta_t^2+2\eta_t D_\Psi(0,w_t)-2\eta_{t+1}D_\Psi(0,w_{t+1}),
\end{aligned}
$$

where we have used $\eta_t \leq (2A)^{-1}\sigma_\Psi, \eta_{t+1} \leq \eta_t$ in the second inequality and the definition of $C_1$ in the last inequality. Taking a summation of the above inequality further implies

$$
\sum_{k=1}^t \eta_k^2\big(f(w_k,z_k)+r(w_k)\big) \leq 2C_1\sum_{k=1}^t \eta_k^2+2\eta_1 D_\Psi(0,w_1) = 2C_1\sum_{k=1}^t \eta_k^2,
$$

where the last identity is due to $w_1 = 0$. This proves (3.6).

We now prove (3.7). Plugging the inequality $\eta_t \leq (2A)^{-1}\sigma_\Psi$ into (B.4) and multiplying both sides by $2\eta_t^{-1}$, we know

$$
f(w_t,z_t)+r(w_t) \leq 2\big(f(0,z_t)+r(0)\big)+2A^{-1}B+2\eta_t^{-1}\big(D_\Psi(0,w_t)-D_\Psi(0,w_{t+1})\big).
$$

Taking a summation of the above inequality, we derive

$$
\sum_{k=1}^t \big(f(w_k,z_k)+r(w_k)\big) \leq 2\sum_{k=1}^t \big(f(0,z_k)+r(0)+A^{-1}B\big)+2\sum_{k=1}^t \eta_k^{-1}\big(D_\Psi(0,w_k)-D_\Psi(0,w_{k+1})\big).
$$

The last term can be controlled by (note $w_1 = 0$)

$$
\sum_{k=1}^t \eta_k^{-1}\big(D_\Psi(0,w_k)-D_\Psi(0,w_{k+1})\big) = \sum_{k=2}^t D_\Psi(0,w_k)\big(\eta_k^{-1}-\eta_{k-1}^{-1}\big)+\eta_1^{-1}D_\Psi(0,w_1)-\eta_t^{-1}D_\Psi(0,w_{t+1})
$$

$$
\leq \max_{1\leq \tilde{k}\leq t} D_\Psi(0,w_{\tilde{k}})\sum_{k=2}^t \big(\eta_k^{-1}-\eta_{k-1}^{-1}\big) \leq \max_{1\leq \tilde{k}\leq t} D_\Psi(0,w_{\tilde{k}})\eta_t^{-1} \leq C_1\Big(\sum_{k=1}^t \eta_k\Big)\eta_t^{-1},
$$

where the last inequality is due to (B.3). Combining the above two inequalities together and using the definition of $C_1$, we derive the stated inequality (3.7). The proof is complete.

$\square$

## C  Proofs for General Convex Objectives

In this section, we prove Theorem 3 and Theorem 4. We first provide a proposition to show that $\|w_{t+1}-w^*\|^2$ can be controlled by $O\big(\sum_{k=1}^t \eta_k^2\|w_k-w^*\|^2\big)$ with high probability. To this aim, we take $w = w^*$ in (3.4) to derive

$$
D_\Psi(w^*,w_{t+1}) \leq \sum_{k=1}^t \xi_k + \sum_{k=1}^t \eta_k\big(\phi(w^*)-\phi(w_k)\big)+\widetilde{C}_1\sum_{k=1}^t \eta_k^2, \tag{C.1}
$$

where $\xi_k$ is defined in (C.4) and $\widetilde{C}_1 \in \mathbb{R}$. A key idea is to use a conditional Bernstein inequality to show $\sum_{k=1}^t \xi_k \leq \sum_{k=1}^t \eta_k\big(\phi(w_k)-\phi(w^*)\big)+\widetilde{C}_2\sum_{k=1}^t \eta_k^2\|w_k-w^*\|^2$ with high probability. An interesting observation is that one can offset the term $\sum_{k=1}^t \eta_k\big(\phi(w^*)-\phi(w_k)\big)$ in (C.1) by the above bound on $\sum_{k=1}^t \xi_k$, leading to the inequality $D_\Psi(w^*,w_{t+1}) \leq \widetilde{C}_1\sum_{k=1}^t \eta_k^2+\widetilde{C}_2\sum_{k=1}^t \eta_k^2\|w_k-w^*\|^2$ with high probability. In the discussion of the conditional variance, we use $\mathbb{E}_{z_k}\big[\big(\xi_k-\mathbb{E}_{z_k}[\xi_k]\big)^2\big] \leq \eta_k^2\|w_k-w^*\|^2(A\phi(w_k)+B)$ and introduce the following decomposition

$$
\eta_k^2\|w_k-w^*\|^2\big(A\phi(w_k)+B\big) = \eta_k^2\|w_k-w^*\|^2 A\big(\phi(w_k)-\phi(w^*)\big)+\eta_k^2\|w_k-w^*\|^2\big(A\phi(w^*)+B\big).
$$

We apply (3.5) to control the first $\|w_k-w^*\|^2$ on the right-hand side to show $\eta_k^2\|w_k-w^*\|^2 A\big(\phi(w_k)-\phi(w^*)\big) \leq \widetilde{C}_3\eta_k\big(\phi(w_k)-\phi(w^*)\big)$ for a $\widetilde{C}_3 > 0$. As a comparison, the second $\|w_k-w^*\|^2$ is kept intact.

**Proposition C.1.** *Let $\{w_t\}_{t\in\mathbb{N}}$ be the sequence produced by (2.2) with $\eta_t \le (2A)^{-1}\sigma_\Psi$ and $\eta_{t+1} \le \eta_t$. We assume $C_6 = \sup_{k\in\mathbb{N}} \eta_k \sum_{j=1}^{k-1} \eta_j < \infty$. Then for any $\delta \in (0,1)$, with probability at least $1-\delta$ we have*

$$\|w_{t+1}-w^*\|^2 \le \frac{A\phi(w^*)+B}{2C_1C_6A} \sum_{k=1}^{t} \eta_k^2\|w_k-w^*\|^2 + \frac{2D_\Psi(w^*,0)}{\sigma_\Psi} + \frac{2C_7 \log\frac{1}{\delta}}{\rho_1\sigma_\Psi} + 4\sigma_\Psi^{-2}(B+AC_1) \sum_{k=1}^{t} \eta_k^2,$$
(C.2)

*where $\rho_1 = \min\{1, (2A)^{-1}(\eta_1\|w^*\|^2 + 2C_1C_6\sigma_\Psi^{-1})^{-1}C_7\}$ and*

$$C_7 = \eta_1\Big(\sup_{z\in\mathcal{Z}} f(w^*,z) + \|w^*\|^2 + AF(0) + B\Big) + 2(A^2+1)C_1\sigma_\Psi^{-1}C_6.$$

*Proof.* Setting $w = w^*$ in (3.4) shows

$$D_\Psi(w^*,w_{t+1}) - D_\Psi(w^*,w_t) \le \eta_t\langle w^* - w_t, f'(w_t,z_t)\rangle + \eta_t(r(w^*) - r(w_t))$$
$$+ \sigma_\Psi^{-1}\eta_t^2\big(Af(w_t,z_t) + Ar(w_t) + 2B\big).$$

We write

$$\langle w^* - w_t, f'(w_t,z_t)\rangle = \langle w^* - w_t, f'(w_t,z_t) - \mathbb{E}_{z_t}[f'(w_t,z_t)]\rangle + \langle w^* - w_t, \mathbb{E}_{z_t}[f'(w_t,z_t)]\rangle$$
$$\le \langle w^* - w_t, f'(w_t,z_t) - \mathbb{E}_{z_t}[f'(w_t,z_t)]\rangle + \big(F(w^*) - F(w_t)\big).$$

Combining the above equations together and using the definition of $\phi$, we derive

$$D_\Psi(w^*,w_{t+1}) - D_\Psi(w^*,w_t) \le \eta_t\langle w^* - w_t, f'(w_t,z_t) - \mathbb{E}_{z_t}[f'(w_t,z_t)]\rangle$$
$$+ \eta_t(\phi(w^*) - \phi(w_t)) + \sigma_\Psi^{-1}\eta_t^2\big(Af(w_t,z_t) + Ar(w_t) + 2B\big).$$

Together with $w_1 = 0$, it then follows that

$$D_\Psi(w^*,w_{t+1}) = D_\Psi(w^*,w_1) + \sum_{k=1}^{t}\big(D_\Psi(w^*,w_{k+1}) - D_\Psi(w^*,w_k)\big)$$

$$\le D_\Psi(w^*,0) + \sum_{k=1}^{t}\eta_k\langle w^* - w_k, f'(w_k,z_k) - \mathbb{E}_{z_k}[f'(w_k,z_k)]\rangle$$

$$+ \sum_{k=1}^{t}\eta_k\big(\phi(w^*) - \phi(w_k)\big) + \sigma_\Psi^{-1}\sum_{k=1}^{t}\eta_k^2\big(Af(w_k,z_k) + Ar(w_k) + 2B\big). \quad \text{(C.3)}$$

Introduce a sequence of random variables as follows

$$\xi_k = \eta_k\langle w^* - w_k, f'(w_k,z_k) - \mathbb{E}_{z_k}[f'(w_k,z_k)]\rangle, \quad k \in \mathbb{N}. \quad \text{(C.4)}$$

It is clear that $\mathbb{E}_{z_k}[\xi_k] = 0$ and therefore $\{\xi_k\}_k$ is a martingale difference sequence. Since $\mathbb{E}[(\xi - \mathbb{E}[\xi])^2] \le \mathbb{E}[\xi^2]$ for any real-valued random variable $\xi$, we know

$$\mathbb{E}_{z_k}\big[|\langle w^* - w_k, f'(w_k,z_k) - \mathbb{E}_{z_k}[f'(w_k,z_k)]\rangle|^2\big] \le \mathbb{E}_{z_k}\big[|\langle w^* - w_k, f'(w_k,z_k)\rangle|^2\big]$$
$$\le \|w^* - w_k\|^2\mathbb{E}_{z_k}\big[\|f'(w_k,z_k)\|_*^2\big] \le \|w^* - w_k\|^2\mathbb{E}_{z_k}\big[Af(w_k,z_k) + B\big]$$
$$\le \|w^* - w_k\|^2\big(AF(w_k) + Ar(w_k) + B\big),$$

where we have used (3.1) in the third inequality. Then, the conditional variance of $\xi_k$ can be controlled by

$$\sum_{k=1}^{t} \mathbb{E}_{z_k}\left[\left(\xi_k - \mathbb{E}_{z_k}[\xi_k]\right)^2\right] = \sum_{k=1}^{t} \eta_k^2 \mathbb{E}_{z_k}\left[\left|\langle w^* - w_k, f'(w_k, z_k) - \mathbb{E}_{z_k}[f'(w_k, z_k)]\rangle\right|^2\right]$$

$$\leq \sum_{k=1}^{t} \eta_k^2 \|w^* - w_k\|^2 \left(A\phi(w_k) - A\phi(w^*)\right) + \sum_{k=1}^{t} \eta_k^2 \|w^* - w_k\|^2 \left(A\phi(w^*) + B\right)$$

$$\leq 2\sum_{k=1}^{t} \eta_k^2 (\|w^*\|^2 + \|w_k\|^2)\left(A\phi(w_k) - A\phi(w^*)\right) + \sum_{k=1}^{t} \eta_k^2 \|w^* - w_k\|^2 \left(A\phi(w^*) + B\right)$$

$$\leq 2A\sum_{k=1}^{t} \eta_k \left(\eta_k\|w^*\|^2 + 2C_1\sigma_\Psi^{-1}\eta_k \sum_{j=1}^{k-1} \eta_j\right)\left(\phi(w_k) - \phi(w^*)\right) + \sum_{k=1}^{t} \eta_k^2 \|w_k - w^*\|^2 \left(A\phi(w^*) + B\right)$$

$$\leq 2A(\eta_1\|w^*\|^2 + 2C_1\sigma_\Psi^{-1}C_6)\sum_{k=1}^{t} \eta_k \left(\phi(w_k) - \phi(w^*)\right) + \sum_{k=1}^{t} \eta_k^2 \|w_k - w^*\|^2 \left(A\phi(w^*) + B\right),$$
(C.5)

where the last second inequality is due to (3.5) and the last inequality is due to the definition of $C_6$.

Furthermore, it follows from the convexity of $f$ that

$$\xi_k - \mathbb{E}_{z_k}[\xi_k] = \eta_k\langle w^* - w_k, f'(w_k, z_k)\rangle + \eta_k\langle w_k - w^*, \mathbb{E}_{z_k}[f'(w_k, z_k)]\rangle$$

$$\leq \eta_k(f(w^*, z_k) - f(w_k, z_k)) + \eta_k\|w_k - w^*\|\left\|\mathbb{E}_{z_k}[f'(w_k, z_k)]\right\|_*. \qquad \text{(C.6)}$$

By the Schwarz's inequality and Lemma A.3, we know

$$\|w_k - w^*\|\left\|\mathbb{E}_{z_k}[f'(w_k, z_k)]\right\|_*$$

$$\leq \frac{1}{2}\left(\|w_k - w^*\|^2 + \|F'(w_k)\|_*^2\right) \leq \frac{1}{2}\left(2\|w_k\|^2 + 2\|w^*\|^2 + 2A^2\|w_k\|^2 + 2AF(0) + 2B\right)$$

$$\leq 2(A^2 + 1)C_1\sigma_\Psi^{-1}\sum_{j=1}^{k-1} \eta_j + \|w^*\|^2 + AF(0) + B,$$

where the last inequality is due to (3.5). Plugging the above inequality back into (C.6) and using the non-negativity of $f(w_t, z_t)$ then give

$$\xi_k - \mathbb{E}_{z_k}[\xi_k] \leq \eta_1\left(\sup_{z\in\mathcal{Z}} f(w^*, z) + \|w^*\|^2 + AF(0) + B\right) + 2(A^2 + 1)C_1\sigma_\Psi^{-1}\eta_k \sum_{j=1}^{k-1} \eta_j \leq C_7.$$

Applying Part (b) of Lemma A.1 with the above estimates on magnitudes and variances of $\xi_k$, we derive the following inequality with probability at least $1 - \delta$

$$\sum_{k=1}^{t} \xi_k \leq \frac{\rho_1}{C_7}\left(2A(\eta_1\|w^*\|^2 + 2C_1\sigma_\Psi^{-1}C_6)\sum_{k=1}^{t} \eta_k \left(\phi(w_k) - \phi(w^*)\right) + \sum_{k=1}^{t} \eta_k^2 \|w_k - w^*\|^2 \left(A\phi(w^*) + B\right)\right)$$

$$+ \frac{C_7\log\frac{1}{\delta}}{\rho_1} \leq \sum_{k=1}^{t} \eta_k \left(\phi(w_k) - \phi(w^*)\right) + \frac{\sigma_\Psi\left(A\phi(w^*) + B\right)}{4C_1C_6A}\sum_{k=1}^{t} \eta_k^2 \|w_k - w^*\|^2 + \frac{C_7\log\frac{1}{\delta}}{\rho_1},$$

where we have used $2\rho_1 A(\eta_1\|w^*\|^2 + 2C_1C_6\sigma_\Psi^{-1}) \leq C_7$. By (3.6) we know

$$\sum_{k=1}^{t} \eta_k^2\left(Af(w_k, z_k) + Ar(w_k) + 2B\right) \leq \left(2AC_1 + 2B\right)\sum_{k=1}^{t} \eta_k^2.$$

Plugging the above two inequalities back into (C.3) gives the following inequality with probability $1 - \delta$

$$D_\Psi(w^*, w_{t+1}) \leq D_\Psi(w^*, 0) + \frac{\sigma_\Psi(A\phi(w^*) + B)}{4C_1C_6A}\sum_{k=1}^{t} \eta_k^2\|w_k - w^*\|^2 + \frac{C_7\log\frac{1}{\delta}}{\rho_1} + 2\sigma_\Psi^{-1}(B + AC_1)\sum_{k=1}^{t} \eta_k^2.$$

This together with the $\sigma_\Psi$-strong convexity of $\Psi$ gives the stated bound with probability $1 - \delta$. The proof is complete.

$\square$

We can use the assumption $\sum_{k=1}^{\infty} \eta_k^2 < \infty$ to show that the right-hand side of (C.2) can be bounded by $\frac{1}{2} \max_{1 \leq k \leq t} \|w_k - w^*\|^2 + \widetilde{C} \log \frac{1}{\delta}$ for a $\widetilde{C} > 0$, from which we can show the boundedness of $\|w_t\|^2$ with high probability (up to a logarithmic factor).

*Proof of Theorem 3.* It follows from the assumption $\sum_{t=1}^{\infty} \eta_t^2 < \infty$ and $\eta_{t+1} \leq \eta_t$ that $\sup_t \eta_t \sum_{j=1}^{t-1} \eta_j \leq \sum_{j=1}^{\infty} \eta_j^2 < \infty$. Therefore, $C_6 = \sup_{k \in \mathbb{N}} \eta_k \sum_{j=1}^{k-1} \eta_j$ is well defined. We define the set $\Omega_T$ as

$$\Omega_T = \left\{ (z_1, \ldots, z_T) : \|w_{t+1} - w^*\|^2 \leq \frac{A\phi(w^*) + B}{2C_1 C_6 A} \sum_{k=1}^{t} \eta_k^2 \|w_k - w^*\|^2 + \frac{2D_\Psi(w^*, 0)}{\sigma_\Psi} \right.$$

$$\left. + \frac{2C_7 \log \frac{T}{\delta}}{\rho_1 \sigma_\Psi} + 4\sigma_\Psi^{-2}(B + AC_1) \sum_{k=1}^{t} \eta_k^2 \text{ for all } t = 1, \ldots, T \right\},$$

where $\rho_1$ is defined in Proposition C.1. By Proposition C.1 and union bounds on probability of events, we know $\Pr\{\Omega_T\} \geq 1 - \delta$. Since $\sum_{t=1}^{\infty} \eta_t^2 < \infty$, we can find a $t_1 \in \mathbb{N}$ such that $(A\phi(w^*) + B) \sum_{k=t_1+1}^{\infty} \eta_k^2 \leq C_1 C_6 A$. With the occurrence of $\Omega_T$, the following inequality holds for all $t = 1, \ldots, T$

$$\|w_{t+1} - w^*\|^2 - \frac{2C_7 \log \frac{T}{\delta}}{\rho_1 \sigma_\Psi} - \frac{2D_\Psi(w^*, 0)}{\sigma_\Psi}$$

$$\leq \frac{A\phi(w^*) + B}{2C_1 C_6 A} \left( \sum_{k=1}^{t_1} \eta_k^2 \|w_k - w^*\|^2 + \sum_{k=t_1+1}^{t} \eta_k^2 \|w_k - w^*\|^2 \right) + \frac{4(B + AC_1) \sum_{k=1}^{t} \eta_k^2}{\sigma_\Psi^2}$$

$$\leq \frac{A\phi(w^*) + B}{2C_1 C_6 A} \left( \sum_{k=1}^{t_1} \eta_k^2 \|w_k - w^*\|^2 + \sum_{k=t_1+1}^{t} \eta_k^2 \sup_{1 \leq \tilde{k} \leq T} \|w_{\tilde{k}} - w^*\|^2 \right) + \frac{4(B + AC_1) \sum_{k=1}^{t} \eta_k^2}{\sigma_\Psi^2}$$

$$\leq \frac{A\phi(w^*) + B}{C_1 C_6 A} \sum_{k=1}^{t_1} \left( 2C_1 \sigma_\Psi^{-1} \eta_k^2 \sum_{j=1}^{k-1} \eta_j + \|w^*\|^2 \right) + \frac{1}{2} \sup_{1 \leq k \leq T} \|w_k - w^*\|^2 + \frac{4(B + AC_1) \sum_{k=1}^{t} \eta_k^2}{\sigma_\Psi^2},$$

where we have used $\|w_k - w^*\|^2 \leq 2(\|w_k\|^2 + \|w^*\|^2)$ and (3.5) in the last step. Under the event $\Omega_T$, we then have

$$\max_{1 \leq t \leq T} \|w_t - w^*\|^2 \leq \frac{A\phi(w^*) + B}{C_1 C_6 A} \sum_{k=1}^{t_1} \left( 2C_1 \sigma_\Psi^{-1} \eta_k^2 \sum_{j=1}^{k-1} \eta_j + \|w^*\|^2 \right)$$

$$+ \frac{1}{2} \sup_{1 \leq k \leq T} \|w_k - w^*\|^2 + \frac{2C_7 \log \frac{T}{\delta}}{\rho_1 \sigma_\Psi} + \frac{2D_\Psi(w^*, 0)}{\sigma_\Psi} + \frac{4(B + AC_1) \sum_{k=1}^{t} \eta_k^2}{\sigma_\Psi^2},$$

from which and $\Pr\{\Omega_T\} \geq 1 - \delta$ we derive the following inequality with probability at least $1 - \delta$

$$\max_{1 \leq t \leq T} \|w_t - w^*\|^2 \leq \frac{2(A\phi(w^*) + B)}{C_1 C_6 A} \sum_{k=1}^{t_1} \left( 2C_1 \sigma_\Psi^{-1} \eta_k^2 \sum_{j=1}^{k-1} \eta_j + \|w^*\|^2 \right)$$

$$+ \frac{4C_7 \log \frac{T}{\delta}}{\rho_1 \sigma_\Psi} + \frac{4D_\Psi(w^*, 0)}{\sigma_\Psi^2} + \frac{8(B + AC_1) \sum_{k=1}^{t} \eta_k^2}{\sigma_\Psi}.$$

The stated inequality holds with $C_2$ defined by (using $(a + b)^2 \leq 2a^2 + 2b^2$)

$$C_2 = \frac{4(A\phi(w^*) + B)}{C_1 C_6 A} \sum_{k=1}^{t_1} \left( 2C_1 \sigma_\Psi^{-1} \eta_k^2 \sum_{j=1}^{k-1} \eta_j + \|w^*\|^2 \right) + \frac{8C_7}{\rho_1 \sigma_\Psi} +$$

$$+ \frac{8D_\Psi(w^*, 0)}{\sigma_\Psi^2} + \frac{16(B + AC_1) \sum_{k=1}^{t} \eta_k^2}{\sigma_\Psi^2} + 2\|w^*\|^2.$$

The proof is complete. $\qquad\square$

We are now in a position to prove Theorem 4. The basic idea is to control $\sum_{t=1}^{T} \eta_t \big(\phi(w_t) - \phi(w^*)\big)$ in terms of a martingale, which can be further controlled by the Azuma-Hoeffding inequality. The bound of $\|w_t\|^2$ in Theorem 3 allows us to control the increments of martingale by logarithmic functions of $T/\delta$.

*Proof of Theorem 4.* It follows from (3.4) that

$$D_\Psi(w, w_{t+1}) - D_\Psi(w, w_t)$$
$$\leq \eta_t \langle w - w_t, f'(w_t, z_t) \rangle + \eta_t \big(r(w) - r(w_t)\big) + \sigma_\Psi^{-1} \eta_t^2 \big(Af(w_t, z_t) + Ar(w_t) + 2B\big)$$
$$\leq \eta_t \langle w - w_t, f'(w_t, z_t) - \mathbb{E}_{z_t}[f'(w_t, z_t)] \rangle + \eta_t \big(\phi(w) - \phi(w_t)\big) + \sigma_\Psi^{-1} \eta_t^2 \big(Af(w_t, z_t) + Ar(w_t) + 2B\big),$$

where we have used the inequality $\langle w - w_t, \mathbb{E}_{z_t}[f'(w_t, z_t)] \rangle \leq F(w) - F(w_t)$ and the definition of $\phi$ in the last inequality.

Taking a summation over $t = 1, \ldots, T$ followed with a reformulation, we derive

$$\sum_{t=1}^{T} \eta_t \big(\phi(w_t) - \phi(w)\big)$$
$$\leq \sum_{t=1}^{T} \xi_t + \sum_{t=1}^{T} \big(D_\Psi(w, w_t) - D_\Psi(w, w_{t+1})\big) + \sigma_\Psi^{-1} \sum_{t=1}^{T} \eta_t^2 \big(Af(w_t, z_t) + Ar(w_t) + 2B\big)$$
$$\leq \sum_{t=1}^{T} \xi_t + D_\Psi(w, 0) + 2\sigma_\Psi^{-1}(AC_1 + B) \sum_{t=1}^{T} \eta_t^2, \qquad (C.7)$$

where we have introduced a sequence of random variables

$$\xi_t = \eta_t \langle w - w_t, f'(w_t, z_t) - \mathbb{E}_{z_t}[f'(w_t, z_t)] \rangle$$

and used (3.6). Let

$$\xi_t' = \eta_t \langle w - w_t, f'(w_t, z_t) - \mathbb{E}_{z_t}[f'(w_t, z_t)] \rangle \mathbb{I}_{\{\|w_t\|^2 \leq C_2 \log \frac{2T}{\delta}\}}, \quad t = 1, \ldots, T,$$

where $\mathbb{I}_\mathcal{A}$ denotes the indicator function of an event $\mathcal{A}$, i.e., $\mathbb{I}_\mathcal{A} = 1$ if $\mathcal{A}$ happens and 0 otherwise. According to the elementary inequality $(a + b)^2 \leq 2(a^2 + b^2)$ for $a, b \in \mathbb{R}$

$$|\xi_t'| \leq \frac{\eta_t}{2} \Big[\|w - w_t\|^2 + \|f'(w_t, z_t) - \mathbb{E}_{z_t}[f'(w_t, z_t)]\|_*^2\Big] \mathbb{I}_{\{\|w_t\|^2 \leq C_2 \log \frac{2T}{\delta}\}}$$
$$\leq \eta_t \Big[\|w\|^2 + \|w_t\|^2 + \|f'(w_t, z_t)\|_*^2 + \|\mathbb{E}_{z_t}[f'(w_t, z_t)]\|_*^2\Big] \mathbb{I}_{\{\|w_t\|^2 \leq C_2 \log \frac{2T}{\delta}\}}.$$

It follows from Lemma A.3 that

$$\|f'(w_t, z_t)\|_*^2 + \|F'(w_t)\|_*^2 \leq 2A^2\|w_t\|^2 + 2Af(0, z_t) + 2B + 2A^2\|w_t\|^2 + 2AF(0) + 2B$$
$$\leq 4A^2\|w_t\|^2 + 2A\big(\sup_z f(0, z) + F(0)\big) + 4B$$
$$\leq 4A^2\|w_t\|^2 + 4AC_1. \qquad (C.8)$$

Combining the above two inequalities together, we derive

$$|\xi_t'| \leq \eta_t \Big[\|w\|^2 + (4A^2 + 1)\|w_t\|^2 + 4AC_1\Big] \mathbb{I}_{\{\|w_t\|^2 \leq C_2 \log \frac{2T}{\delta}\}}$$
$$\leq \eta_t \Big(\|w\|^2 + 4AC_1 + (4A^2 + 1)C_2 \log \frac{2T}{\delta}\Big) \leq C(w)\eta_t \log \frac{2T}{\delta},$$

where we introduce $C(w) = \|w\|^2 + 4AC_1 + (4A^2 + 1)C_2$. It is clear that $\mathbb{E}_{z_t}[\xi_t'] = 0$ and $\xi_t'$ depends only on $z_1, \ldots, z_t$. According to Part (a) of Lemma A.1, we can find an event $\Omega_T := \{(z_1, \ldots, z_T) : z_1, \ldots, z_T \in \mathcal{Z}\}$ with $\Pr\{\Omega_t\} \geq 1 - \frac{\delta}{2}$ such that for any $(z_1, \ldots, z_T) \in \Omega_T$ the following inequality holds

$$\sum_{t=1}^{T} \xi_t' \leq C(w) \log \frac{2T}{\delta} \Big(2 \sum_{t=1}^{T} \eta_t^2 \log \frac{2}{\delta}\Big)^{\frac{1}{2}} \leq C(w) \log^{\frac{3}{2}} \frac{2T}{\delta} \Big(2 \sum_{t=1}^{T} \eta_t^2\Big)^{\frac{1}{2}}.$$

Furthermore, according to Theorem 3, there exists an event $\Omega'_T := \{(z_1, \ldots, z_T) : z_1, \ldots, z_T \in \mathcal{Z}\}$ with $\Pr\{\Omega'_t\} \geq 1 - \frac{\delta}{2}$ such that for any $(z_1, \ldots, z_T) \in \Omega'_T$ the following inequality holds

$$\max_{1 \leq t \leq T} \|w_t\|^2 \leq C_2 \log \frac{2T}{\delta}.$$

Under the intersection of these two events, we have $\xi_t = \xi'_t$ and therefore

$$\sum_{t=1}^T \xi_t = \sum_{t=1}^T \xi'_t \leq C(w) \log^{\frac{3}{2}} \frac{2T}{\delta} \Big(2 \sum_{t=1}^T \eta_t^2\Big)^{\frac{1}{2}},$$

which, together with $\Pr\{\Omega_T \cap \Omega'_T\} \geq 1 - \delta$ and (C.7), shows the following inequality with probability at least $1 - \delta$

$$\sum_{t=1}^T \eta_t \big(\phi(w_t) - \phi(w)\big) \leq D_\Psi(w, 0) + 2\sigma_\Psi^{-1}\big(AC_1 + B\big) \sum_{t=1}^T \eta_t^2 + C(w) \log^{\frac{3}{2}} \frac{2T}{\delta} \Big(2 \sum_{t=1}^T \eta_t^2\Big)^{\frac{1}{2}}$$

$$\leq \big(2C_3 D_\Psi(w, 0) + C_4\big) \log^{\frac{3}{2}} \frac{2T}{\delta},$$

where

$$C_3 = 2^{-1} + \sigma_\Psi^{-1}\Big(2 \sum_{t=1}^\infty \eta_t^2\Big)^{\frac{1}{2}} \quad \text{and} \quad C_4 := 2\sigma_\Psi^{-1}(AC_1 + B) \sum_{t=1}^\infty \eta_t^2 + (4AC_1 + 4A^2 C_2 + C_2)\Big(2 \sum_{t=1}^\infty \eta_t^2\Big)^{\frac{1}{2}}.$$

The stated inequality then follows from the convexity of $\phi$. The proof is complete. $\qquad\square$

# D   Proofs for Strongly Convex Objectives

This section is devoted to proving Theorem 8. First, we take a weighted summation of (3.4) and use (3.7) to tackle $\sum_{k=1}^t \big(f(w_k, z_k) + r(w_k)\big)$ without boundedness assumptions, yielding Lemma D.2. We need the following simple lemma on step sizes in this derivation.

**Lemma D.1.** *Let* $\eta_k = \frac{2}{\sigma_\phi k + 2\sigma_F + \sigma_\phi t_0}$, *where* $t_0 \in \mathbb{R}_+$. *Then,*

$$\sum_{k=1}^t \eta_k \leq 2\sigma_\phi^{-1} \log(et). \tag{D.1}$$

*Proof.* It follows from the definition of $\eta_t$ that

$$\sum_{k=1}^t \eta_k \leq 2\sigma_\phi^{-1} \sum_{k=1}^t (k + t_0)^{-1} \leq 2\sigma_\phi^{-1} \log(et).$$

The proof is complete. $\qquad\square$

**Lemma D.2.** *Assume* $\sigma_\phi > 0$. *Let* $\{w_t\}_{t \in \mathbb{N}}$ *be the sequence produced by (2.2) with* $\eta_t = \frac{2}{\sigma_\phi t + 2\sigma_F + \sigma_\phi t_0}$, *where* $t_0 \geq 4A/(\sigma_\Psi \sigma_\phi)$. *Then the following inequality holds for all* $t = 1, \ldots, T$

$$2\sigma_\phi^{-1} \sum_{k=1}^t (k + t_0 + 1)\big(\phi(w_k) - \phi(w^*)\big) + (t + t_0 + 1)(t + t_0 + 2)D_\Psi(w^*, w_{t+1}) \leq (t_0 + 1)(t_0 + 2)D_\Psi(w^*, w_1)$$

$$+ 2\sigma_\phi^{-1} \sum_{k=1}^t (k + t_0 + 1)\xi_k + 16 \log(eT)\sigma_\Psi^{-1}\sigma_\phi^{-2}\big(AC_1(2t + t_0 + 2) + Bt\big), \quad \text{(D.2)}$$

*where we introduce*

$$\xi_k = \langle w^* - w_k, f'(w_k, z_k) - \mathbb{E}_{z_k}[f'(w_k, z_k)]\rangle, \quad k = 1, \ldots, T.$$

*Proof.* Since $t_0 \geq \frac{4A}{\sigma_\Psi \sigma_\phi}$, we know $\eta_t \leq (2A)^{-1}\sigma_\Psi$ and therefore Lemma 2 holds. Taking $w = w^*$ in (3.4), we derive

$$D_\Psi(w^*, w_{k+1}) - D_\Psi(w^*, w_k) \leq \eta_k \langle w^* - w_k, f'(w_k, z_k) - \mathbb{E}_{z_k}[f'(w_k, z_k)] \rangle + \eta_k \langle w^* - w_k, F'(w_k) \rangle$$
$$+ \eta_k \big( r(w^*) - r(w_k) \big) + \sigma_\Psi^{-1}\eta_k^2 \big( Af(w_k, z_k) + Ar(w_k) + 2B \big) - \sigma_r \eta_k D_\Psi(w^*, w_{k+1}).$$

Plugging the inequality $F(w^*) - F(w_k) \geq \langle w^* - w_k, F'(w_k) \rangle + \sigma_F D_\Psi(w^*, w_k)$ into the above inequality then shows

$$D_\Psi(w^*, w_{k+1}) - D_\Psi(w^*, w_k) \leq \eta_k \xi_k + \eta_k \big( F(w^*) - F(w_k) - \sigma_F D_\Psi(w^*, w_k) \big)$$
$$+ \eta_k \big( r(w^*) - r(w_k) \big) + \sigma_\Psi^{-1}\eta_k^2 \big( Af(w_k, z_k) + Ar(w_k) + 2B \big) - \sigma_r \eta_k D_\Psi(w^*, w_{k+1}).$$

According to the definition of $\phi$, we further get

$$(1 + \sigma_r \eta_k) D_\Psi(w^*, w_{k+1}) \leq (1 - \eta_k \sigma_F) D_\Psi(w^*, w_k) + \eta_k \xi_k + \eta_k \big( \phi(w^*) - \phi(w_k) \big)$$
$$+ \sigma_\Psi^{-1}\eta_k^2 \big( Af(w_k, z_k) + Ar(w_k) + 2B \big), \quad \text{(D.3)}$$

which can be reformulated as follows

$$\frac{\eta_k \big( \phi(w_k) - \phi(w^*) \big)}{1 + \sigma_r \eta_k} + D_\Psi(w^*, w_{k+1}) \leq \frac{1 - \eta_k \sigma_F}{1 + \eta_k \sigma_r} D_\Psi(w^*, w_k) + \frac{\eta_k \xi_k}{1 + \sigma_r \eta_k}$$
$$+ \frac{\sigma_\Psi^{-1}\eta_k^2 \big( Af(w_k, z_k) + Ar(w_k) + 2B \big)}{1 + \sigma_r \eta_k}. \quad \text{(D.4)}$$

Since $\eta_k = \frac{2}{\sigma_\phi k + 2\sigma_F + \sigma_\phi t_0}$, we know

$$\frac{1 - \sigma_F \eta_k}{1 + \sigma_r \eta_k} = \frac{\sigma_\phi k + 2\sigma_F + \sigma_\phi t_0 - 2\sigma_F}{\sigma_\phi k + 2\sigma_F + \sigma_\phi t_0 + 2\sigma_r} = \frac{k + t_0}{k + t_0 + 2},$$

$$\frac{\eta_k}{1 + \sigma_r \eta_k} = \frac{2}{\sigma_\phi (k + t_0 + 2)}.$$

Plugging the above two equations back into (D.4), we derive

$$\frac{2 \big( \phi(w_k) - \phi(w^*) \big)}{\sigma_\phi (k + t_0 + 2)} + D_\Psi(w^*, w_{k+1}) \leq \frac{k + t_0}{k + t_0 + 2} D_\Psi(w^*, w_k) + \frac{2\xi_k}{\sigma_\phi (k + t_0 + 2)}$$
$$+ \frac{2\eta_k \big( Af(w_k, z_k) + Ar(w_k) + 2B \big)}{\sigma_\Psi \sigma_\phi (k + t_0 + 2)}.$$

Multiplying both sides by $(k + t_0 + 1)(k + t_0 + 2)$, we get

$$\frac{2(k + t_0 + 1)}{\sigma_\phi} \big( \phi(w_k) - \phi(w^*) \big) + (k + t_0 + 1)(k + t_0 + 2) D_\Psi(w^*, w_{k+1})$$

$$\leq (k+t_0)(k+t_0+1) D_\Psi(w^*, w_k) + \frac{2(k + t_0 + 1)\xi_k}{\sigma_\phi} + \frac{2\eta_k (k + t_0 + 1) \big( Af(w_k, z_k) + Ar(w_k) + 2B \big)}{\sigma_\Psi \sigma_\phi}.$$

Taking a summation of the above inequality from $k = 1$ to $k = t$ and using the inequality $(k + t_0 + 1)\eta_k \leq 4\sigma_\phi^{-1}$, we derive

$$2\sigma_\phi^{-1} \sum_{k=1}^{t} (k+t_0+1) \big( \phi(w_k) - \phi(w^*) \big) + (t+t_0+1)(t+t_0+2) D_\Psi(w^*, w_{t+1}) \leq (t_0+1)(t_0+2) D_\Psi(w^*, w_1)$$

$$+ 2\sigma_\phi^{-1} \sum_{k=1}^{t} (k + t_0 + 1)\xi_k + 8\sigma_\Psi^{-1}\sigma_\phi^{-2} \sum_{k=1}^{t} \big( Af(w_k, z_k) + Ar(w_k) + 2B \big). \quad \text{(D.5)}$$

According to (3.7), (D.1) and $\eta_t^{-1} \leq 2^{-1}\sigma_\phi (t + t_0 + 2)$, we know

$$\sum_{k=1}^{t} \big( Af(w_k, z_k) + Ar(w_k) + 2B \big) \leq t(2AC_1 + 2B) + 2AC_1 \Big( \sum_{k=1}^{t} \eta_k \Big) \eta_t^{-1}$$

$$\leq 2t(AC_1 + B) + 2AC_1 \Big( 2\sigma_\phi^{-1} \log(et) \Big) \Big( 2^{-1}\sigma_\phi (t + t_0 + 2) \Big)$$
$$= 2t(AC_1 + B) + 2AC_1 (t + t_0 + 2) \log(et)$$
$$\leq 2\log(eT) \big( AC_1(2t + t_0 + 2) + Bt \big).$$

Plugging the above inequality into (D.5) gives the stated inequality. The proof is complete. $\qquad\square$

In the following lemma, we establish bounds on magnitudes and conditional variances on $\{\xi_k\}_k$ defined in Lemma D.2.

**Lemma D.3.** *Let the assumptions of Lemma D.2 hold with $t_0 \geq \frac{4A}{\sigma_\Psi \sigma_\phi}$ and $\xi_k$ be defined in Lemma D.2. Then for all $k \leq T$ we have*

$$|\xi_k| \leq C_8 \log(eT) \quad \text{and} \quad \mathbb{E}_{z_k}\big[\big(\xi_k - \mathbb{E}_{z_k}[\xi_k]\big)^2\big] \leq \|w^* - w_k\|^2 \big(A\phi(w_k) + B\big),$$

*where*

$$C_8 := (16A^2 + 4)C_1 \sigma_\Psi^{-1} \sigma_\phi^{-1} + \|w^*\|^2 + 4AC_1.$$

*Proof.* Since $t_0 \geq \frac{4A}{\sigma_\Psi \sigma_\phi}$, we know $\eta_t \leq (2A)^{-1}\sigma_\Psi$ and therefore (3.5) holds. According to Schwarz's inequality, we have

$$\big|\langle w^* - w_k, f'(w_k, z_k) - \mathbb{E}_{z_k}[f'(w_k, z_k)]\rangle\big| \leq \|w^* - w_k\| \big(\|f'(w_k, z_k)\|_* + \|F'(w_k)\|_*\big)$$
$$\leq \frac{1}{2}\|w^* - w_k\|^2 + \frac{1}{2}\big(\|f'(w_k, z_k)\|_* + \|F'(w_k)\|_*\big)^2$$
$$\leq \|w^*\|^2 + \|w_k\|^2 + \|f'(w_k, z_k)\|_*^2 + \|F'(w_k)\|_*^2.$$

Combining the above inequality and (C.8) together shows

$$\big|\langle w^* - w_k, f'(w_k, z_k) - \mathbb{E}_{z_k}[f'(w_k, z_k)]\rangle\big| \leq (4A^2 + 1)\|w_k\|^2 + \|w^*\|^2 + 4AC_1$$

$$\leq (8A^2 + 2)C_1\sigma_\Psi^{-1} \sum_{j=1}^{k} \eta_j + \|w^*\|^2 + 4AC_1 \leq C_8 \log(ek),$$

where we have used (3.5) and Lemma D.1 to control $\sum_{j=1}^{k} \eta_j$. This shows a bound on $|\xi_k|$.

It is clear that $\mathbb{E}_{z_k}[\xi_k] = 0$ and therefore it follows from $\mathbb{E}[(\xi - \mathbb{E}[\xi])^2] \leq \mathbb{E}[\xi^2]$ for all real-valued random variable $\xi$ that

$$\mathbb{E}_{z_k}\big[\big(\xi_k - \mathbb{E}_{z_k}[\xi_k]\big)^2\big] = \mathbb{E}_{z_k}[\xi_k^2] \leq \mathbb{E}_{z_k}\big[\langle w^* - w_k, f'(w_k, z_k)\rangle^2\big]$$
$$\leq \|w^* - w_k\|^2 \mathbb{E}_{z_k}[\|f'(w_k, z_k)\|_*^2] \leq \|w^* - w_k\|^2 \big(AF(w_k) + B\big)$$
$$\leq \|w^* - w_k\|^2 \big(A\phi(w_k) + B\big),$$

where we have used

$$\mathbb{E}_{z_k}[\|f'(w_k, z_k)\|_*^2] \leq \mathbb{E}_{z_k}[Af(w_k, z_k) + B] = AF(w_k) + B$$

due to (3.1). The proof is complete. $\qquad\square$

Then, we apply a Bernstein inequality to show $\sum_{k=1}^{t}(k + t_0 + 1)\xi_k \leq \frac{1}{2}\sum_{k=1}^{t}(k + t_0 + 1)\big(\phi(w_k) - \phi(w^*)\big) + \mathfrak{C}_t$ with high probability, where $\mathfrak{C}_t$ is the summation of the last two terms in (D.10). An interesting observation is that $\frac{1}{2}\sum_{k=1}^{t}(k + t_0 + 1)\big(\phi(w_k) - \phi(w^*)\big)$ can be offset by the first term in (D.2), from which one can derive (3.10). To apply the Bernstein inequality, we use Lemma D.3 to control the conditional variance as $\mathbb{E}_{z_k}\big[\big(\xi_k - \mathbb{E}_{z_k}[\xi_k]\big)^2\big] \leq \|w^* - w_k\|^2 \big(A\phi(w_k) + B\big)$, and introduce the decomposition $A\phi(w_k) + B = A\phi(w_k) - A\phi(w^*) + A\phi(w^*) + B$ to get variance partially offset by the first term in (D.2). This is a key trick for us to proceed with the discussion without boundedness assumption on subgradients.

*Proof of Theorem 8.* Let $\xi_k$ be defined in Lemma D.2. Since $t_0 \geq \frac{16A \log \frac{T}{\delta}}{\sigma_\phi \sigma_\Psi}$ and $\delta \leq e^{-\frac{1}{4}}$, we know $t_0 \geq \frac{4A}{\sigma_\Psi \sigma_\phi}$ and therefore Lemma D.2 and Lemma D.3 hold. Define

$$C_T = \max\left\{ \frac{4(t_0 + 1)D_\Psi(w^*, w_1)}{\sigma_\Psi} + \frac{3t_0\big(\phi(w^*) + A^{-1}B\big)}{\sigma_\phi \sigma_\Psi} + \right.$$
$$\left. \frac{64 \log(eT)(B + 2AC_1)}{\sigma_\Psi^2 \sigma_\phi^2}, \frac{C_8 t_0 \log(eT)}{2A} \right\}. \quad \text{(D.6)}$$

Let $\rho_2 = \frac{C_8 t_0 \log(eT)}{2AC_T}$. It is clear from the definition of $C_T$ that $\rho_2 \in (0,1]$. According to Lemma D.3, we derive the following inequalities for all $k = 1, \ldots, t (t \leq T)$

$$|(k + t_0 + 1)\xi_k| \leq C_8(t + t_0 + 1)\log(eT)$$

$$\mathbb{E}_{z_k}\left[\left((k + t_0 + 1)\xi_k - \mathbb{E}_{z_k}[(k + t_0 + 1)\xi_k]\right)^2\right] \leq (k + t_0 + 1)^2 \|w^* - w_k\|^2 (A\phi(w_k) + B).$$

Plugging the above two inequalities back into Part (b) of Lemma A.1, we derive the following inequality with probability at least $1 - \frac{\delta}{T}$

$$\sum_{k=1}^{t}(k + t_0 + 1)\xi_k \leq \frac{\rho_2 \sum_{k=1}^{t}\left((k + t_0 + 1)^2 \|w^* - w_k\|^2 (A\phi(w_k) + B)\right)}{C_8(t + t_0 + 1)\log(eT)}$$

$$+ \frac{C_8(t + t_0 + 1)\log(eT)\log\frac{T}{\delta}}{\rho_2}. \quad \text{(D.7)}$$

Taking union bounds on probabilities of events, it is clear that (D.7) holds with probability at least $1 - \delta$ simultaneously for all $t = 1, \ldots, T$. In the remainder of the proof, we always assume that (D.7) holds for all $t = 1, \ldots, T$, which happens with probability at least $1 - \delta$.

Applying the $\sigma_\Psi$-strong convexity of $\Psi$ to (D.2) and dividing both sides by $2^{-1}\sigma_\Psi(t + t_0 + 1)(t + t_0 + 2)$, we derive the following inequality with probability $1 - \delta$ for all $t = 1, \ldots, T$

$$\frac{4\sum_{k=1}^{t}(k + t_0 + 1)\left(\phi(w_k) - \phi(w^*)\right)}{\sigma_\phi\sigma_\Psi(t + t_0 + 1)(t + t_0 + 2)} + \|w^* - w_{t+1}\|^2 \leq \frac{2(t_0 + 1)(t_0 + 2)D_\Psi(w^*, w_1)}{\sigma_\Psi(t + t_0 + 1)(t + t_0 + 2)} +$$

$$\frac{4\sum_{k=1}^{t}(k + t_0 + 1)\xi_k}{(t + t_0 + 1)(t + t_0 + 2)\sigma_\phi\sigma_\Psi} + \frac{32\log(eT)\left(AC_1(2t + t_0 + 2) + Bt\right)}{(t + t_0 + 1)(t + t_0 + 2)\sigma_\Psi^2\sigma_\phi^2}. \quad \text{(D.8)}$$

We now show by induction that $\|w^* - w_{\tilde{t}}\|^2 \leq \frac{C_T}{\tilde{t} + t_0 + 1}$ for all $\tilde{t} = 1, \ldots, T$. It is clear from the definition of $C_T$ that

$$\|w^* - w_1\|^2 \leq \frac{2D_\Psi(w^*, w_1)(t_0 + 2)}{\sigma_\Psi(t_0 + 2)} \leq \frac{4(t_0 + 1)D_\Psi(w^*, w_1)}{\sigma_\Psi(t_0 + 2)} \leq \frac{C_T}{t_0 + 2}.$$

Therefore, the induction assumption holds for the case with $\tilde{t} = 1$. Suppose that $\|w^* - w_{\tilde{t}}\|^2 \leq \frac{C_T}{\tilde{t} + t_0 + 1}$ for all $\tilde{t} \leq t$. We now need to show that it also holds for $\tilde{t} = t + 1$, i.e., $\|w^* - w_{t+1}\|^2 \leq \frac{C_T}{t + t_0 + 2}$. According to (D.8) multiplied by $t + t_0 + 2$, it suffices to show

$$-\frac{4\sum_{k=1}^{t}(k + t_0 + 1)\left(\phi(w_k) - \phi(w^*)\right)}{\sigma_\phi\sigma_\Psi(t + t_0 + 1)} + \frac{2(t_0 + 1)(t_0 + 2)D_\Psi(w^*, w_1)}{\sigma_\Psi(t + t_0 + 1)} +$$

$$\frac{4\sum_{k=1}^{t}(k + t_0 + 1)\xi_k}{\sigma_\phi\sigma_\Psi(t + t_0 + 1)} + \frac{32\log(eT)\left(AC_1(2t + t_0 + 2) + Bt\right)}{\sigma_\Psi^2\sigma_\phi^2(t + t_0 + 1)} \leq C_T. \quad \text{(D.9)}$$

Plugging the induction assumption $\|w^* - w_{\tilde{t}}\|^2 \leq C_T/(\tilde{t} + t_0 + 1)$ for all $\tilde{t} \leq t$ back into (D.7), we derive

$$\sum_{k=1}^{t}(k + t_0 + 1)\xi_k$$

$$\leq \frac{\rho_2 C_T \sum_{k=1}^{t}\left((k + t_0 + 1)\left(A\phi(w_k) + B\right)\right)}{C_8(t + t_0 + 1)\log(eT)} + \frac{C_8(t + t_0 + 1)\log(eT)\log\frac{T}{\delta}}{\rho_2}$$

$$= \frac{t_0 A^{-1}}{2(t + t_0 + 1)}\sum_{k=1}^{t}(k + t_0 + 1)\left(A\phi(w_k) - A\phi(w^*) + A\phi(w^*) + B\right) + \frac{2(t + t_0 + 1)AC_T\log\frac{T}{\delta}}{t_0}$$

$$\leq \frac{1}{2}\sum_{k=1}^{t}(k + t_0 + 1)\left(\phi(w_k) - \phi(w^*)\right) + \frac{t_0(A\phi(w^*) + B)\sum_{k=1}^{t}(k + t_0 + 1)}{2A(t + t_0 + 1)} + \frac{(t + t_0 + 1)C_T\sigma_\phi\sigma_\Psi}{8},$$

$$\text{(D.10)}$$

where we have used the definition of $\rho_2$ in the first identity and the assumption $t_0 \geq \frac{16A \log \frac{T}{\delta}}{\sigma_\phi \sigma_\Psi}$ in the last inequality. Plugging the above inequality into (D.9), it suffices to show

$$\frac{2(t_0+1)(t_0+2)D_\Psi(w^*, w_1)}{\sigma_\Psi(t+t_0+1)} + \frac{2t_0(\phi(w^*)+A^{-1}B)\sum_{k=1}^{t}(k+t_0+1)}{\sigma_\Psi\sigma_\phi(t+t_0+1)^2} + \frac{C_T}{2}$$
$$+ \frac{32(B+2AC_1)\log(eT)}{\sigma_\Psi^2\sigma_\phi^2} \leq C_T.$$

Since

$$\sum_{k=1}^{t}(k+t_0+1) = \frac{t(t+2t_0+3)}{2} \leq \frac{3(t+t_0+1)^2}{4}, \tag{D.11}$$

it suffices to show

$$\frac{2(t_0+1)D_\Psi(w^*, w_1)}{\sigma_\Psi} + \frac{3t_0(\phi(w^*)+A^{-1}B)}{2\sigma_\Psi\sigma_\phi} + \frac{C_T}{2} + \frac{32(B+2AC_1)\log(eT)}{\sigma_\Psi^2\sigma_\phi^2} \leq C_T.$$

which is clear from the definition of $C_T$ in (D.6). Therefore, $\|w^* - w_{t+1}\|^2 \leq \frac{C_T}{t+t_0+2}$. This proves the first inequality in (3.10).

We now prove the second inequality in (3.10). According to (D.2), we know

$$\sum_{k=1}^{t}(k+t_0+1)\big(\phi(w_k)-\phi(w^*)\big) \leq \frac{\sigma_\phi(t_0+1)(t_0+2)D_\Psi(w^*, w_1)}{2} +$$
$$\sum_{k=1}^{t}(k+t_0+1)\xi_k + \frac{8\log(eT)\big(AC_1(2t+t_0+2)+Bt\big)}{\sigma_\phi\sigma_\Psi}.$$

Plugging (D.10) into the above inequality and using (D.11), we derive the following inequality with probability at least $1-\delta$ for all $t = 1, \ldots, T$

$$\frac{\sum_{k=1}^{t}(k+t_0+1)\big(\phi(w_k)-\phi(w^*)\big)}{2} \leq \frac{\sigma_\phi(t_0+1)(t_0+2)D_\Psi(w^*, w_1)}{2} +$$
$$\frac{3t_0\big(A\phi(w^*)+B\big)(t+t_0+1)}{8A} + \frac{(t+t_0+1)C_T\sigma_\phi\sigma_\Psi}{8} + \frac{8\log(eT)\big(AC_1(2t+t_0+2)+Bt\big)}{\sigma_\phi\sigma_\Psi}.$$

With probability at least $1-\delta$, it then follows from the convexity of $\phi$ and the identity in (D.11) that

$$\phi(\bar{w}_t^{(2)}) - \phi(w^*) \leq \Big(\sum_{k=1}^{t}(k+t_0+1)\Big)^{-1}\Big(\sum_{k=1}^{t}(k+t_0+1)\big(\phi(w_k)-\phi(w^*)\big)\Big)$$
$$\leq \frac{1}{t(t+2t_0+3)}\Big(2\sigma_\phi(t_0+1)(t_0+2)D_\Psi(w^*, w_1) + \frac{3t_0\big(A\phi(w^*)+B\big)(t+t_0+1)}{2A}$$
$$+ \frac{(t+t_0+1)C_T\sigma_\phi\sigma_\Psi}{2} + \frac{32\log(eT)\big(AC_1(2t+t_0+2)+Bt\big)}{\sigma_\phi\sigma_\Psi}\Big), \quad \text{for all } t = 1, \ldots, T.$$

This establishes the second inequality in (3.10) with $\widetilde{C}_T$ defined by

$$\widetilde{C}_T = \sigma_\phi(t_0+1)D_\Psi(w^*, w_1) + \frac{3t_0\big(A\phi(w^*)+B\big)}{2A} + \frac{C_T\sigma_\phi\sigma_\Psi}{2} + \frac{32\log(eT)(2AC_1+B)}{\sigma_\phi\sigma_\Psi}.$$

The proof is complete. $\qquad\square$

**Remark 1.** According to the definition of $C_T$ and $\widetilde{C}_T$, it is clear that both $C_T$ and $\widetilde{C}_T$ only involves logarithmic functions of $T/\delta$. It is also clear that $C_T$ is a quadratic function of $\sigma_\phi^{-1}$ and $\widetilde{C}_T$ is a linear function of $\sigma_\phi^{-1}$.

# E   Proofs for Almost Sure Convergence

In this section, we present a proposition on almost sure convergence which covers both the general convex case (Theorem 6) and the strongly convex case (Theorem 9). To this aim, we need to introduce two lemmas. Lemma E.1 is the Doob's martingale convergence theorem [see, e.g., 2, page 195] which is a powerful tool to study almost sure convergence. We will use Lemma E.2 [9] to show that the random variable to which $D_\Psi(w^*, w_t)$ converges is zero almost surely in the strongly convex case.

**Lemma E.1.** *Let $\{\tilde{X}_t\}_{t\in\mathbb{N}}$ be a sequence of non-negative random variables with $\mathbb{E}[\tilde{X}_1] < \infty$ and let $\{\mathcal{F}_t\}_{t\in\mathbb{N}}$ be a nested sequence of sets of random variables with $\mathcal{F}_t \subset \mathcal{F}_{t+1}$ for all $t \in \mathbb{N}$. If $\mathbb{E}[\tilde{X}_{t+1}|\mathcal{F}_t] \le \tilde{X}_t$ for every $t \in \mathbb{N}$, then $\tilde{X}_t$ converges to a nonnegative random variable $\tilde{X}$ almost surely. Furthermore, $\tilde{X} < \infty$ almost surely.*

**Lemma E.2.** *Let $\{\eta_t\}_{t\in\mathbb{N}}$ be a sequence of non-negative numbers such that $\lim_{t\to\infty} \eta_t = 0$ and $\sum_{t=1}^\infty \eta_t = \infty$. Let $a > 0$ and $t_1 \in \mathbb{N}$ such that $\eta_t < a^{-1}$ for any $t \ge t_1$. Then we have $\lim_{T\to\infty} \sum_{t=t_1}^T \eta_t^2 \prod_{k=t+1}^T (1 - a\eta_k) = 0$.*

The basic idea in proving Proposition E.3 is to construct non-negative supermartingales based on the one-step progress inequality (3.4), whose almost sure convergence based on Lemma E.1 will imply the almost sure convergence of the random variables we are interested in. We will construct different supermartingales in the general convex case and the strongly convex case.

**Proposition E.3.** *Let $\{w_t\}_{t\in\mathbb{N}}$ be the sequence produced by (2.2). If $\|w^*\| < \infty$ and $\sum_{t=1}^\infty \eta_t^2 < \infty$, then $\{D_\Psi(w^*, w_t)\}_t$ converges almost surely to a non-negative random variable and $\lim_{t\to\infty} D_\Psi(w^*, w_t) < \infty$ almost surely. Furthermore,*

*(a) if $\eta_t \le (2A)^{-1}\sigma_\Psi$ and $\eta_{t+1} \le \eta_t$, then $\sum_{t=1}^\infty \eta_t\big(\phi(w_t) - \phi(w^*)\big) < \infty$ almost surely;*

*(b) if $\sigma_\phi > 0$ and $\sum_{t=1}^\infty \eta_t = \infty$, then $\lim_{t\to\infty} D_\Psi(w^*, w_t) = 0$ almost surely.*

*Proof.* Since $\sum_{t=1}^\infty \eta_t^2 < \infty$, there exists a $t_2 \in \mathbb{N}$ such that $\eta_t \le \min\{(2A)^{-1}\sigma_\Psi, 2\sigma_\phi^{-1}, \sigma_r^{-1}\}$ for all $t \ge t_2$. Taking conditional expectations w.r.t. $z_t$ on both sides of (D.3), we derive the following inequality for all $t \ge t_2$

$$\mathbb{E}_{z_t}[D_\Psi(w^*, w_{t+1})] \le \frac{1 - \sigma_F \eta_t}{1 + \sigma_r \eta_t} D_\Psi(w^*, w_t) + \frac{\eta_t}{1 + \sigma_r \eta_t}\big(\phi(w^*) - \phi(w_t)\big)$$
$$+ \sigma_\Psi^{-1}\eta_t^2\big(A\phi(w_t) - A\phi(w^*) + A\phi(w^*) + 2B\big),$$

where we have used $1 + \sigma_F \eta_t \ge 1$ and $\mathbb{E}_{z_t}[\xi_t] = 0$ for $\xi_t$ defined in Lemma D.2. According to $\phi(w^*) \le \phi(w_t)$ and $\eta_t \le \min\{(2A)^{-1}\sigma_\Psi, \sigma_r^{-1}\}$, we know

$$\eta_t(1 + \sigma_r \eta_t)^{-1}\big(\phi(w^*) - \phi(w_t)\big) + \sigma_\Psi^{-1}\eta_t^2\big(A\phi(w_t) - A\phi(w^*)\big)$$
$$\le 2^{-1}\eta_t\big(\phi(w^*) - \phi(w_t)\big) + 2^{-1}\eta_t\big(\phi(w_t) - \phi(w^*)\big) = 0.$$

Combining the above two inequalities together, we derive

$$\mathbb{E}_{z_t}[D_\Psi(w^*, w_{t+1})] \le (1 - \sigma_F \eta_t)(1 + \sigma_r \eta_t)^{-1} D_\Psi(w^*, w_t) + \sigma_\Psi^{-1}\eta_t^2\big(A\phi(w^*) + 2B\big). \quad \text{(E.1)}$$

Introduce a sequence of non-negative random variables $\widetilde{X}_t$ as

$$\widetilde{X}_t = D_\Psi(w^*, w_t) + \sigma_\Psi^{-1}\big(A\phi(w^*) + 2B\big)\sum_{k=t}^\infty \eta_k^2,$$

which is well defined since $\sum_{t=1}^\infty \eta_t^2 < \infty$. By (E.1), it is clear that $\mathbb{E}_{z_t}[\widetilde{X}_{t+1}] \le \widetilde{X}_t$ for all $t \ge t_2$. Taking $w = w^*$ and expectations on both sides of (B.2), we derive

$$\mathbb{E}[D_\Psi(w^*, w_{t+1})] \le \mathbb{E}[D_\Psi(w^*, w_t)] + \sigma_\Psi^{-1}\eta_t^2 A\mathbb{E}[\phi(w_t)] + 2\sigma_\Psi^{-1}B\eta_t^2, \quad \text{for all } t \in \mathbb{N},$$

where we have used $\phi(w^*) \le \phi(w_t)$. According to Lemma A.3, the term $\mathbb{E}[\phi(w_t)]$ can be controlled by $\mathbb{E}[D_\Psi(w^*, w_t)]$ and $\|w^*\|$. Therefore, we derive an upper bound on $\mathbb{E}[D_\Psi(w^*, w_{t+1})]$ in terms

of $\mathbb{E}[D_\Psi(w^*, w_t)]$, $\|w^*\|$ and step sizes, from which we know $\mathbb{E}[\widetilde{X}_{t_2}] < \infty$ ($t_2$ is a fixed constant). Therefore, one can apply Lemma E.1 to show that $\widetilde{X}_t$ converges almost surely to a non-negative random variable, which, together with $\sum_{t=1}^\infty \eta_t^2 < \infty$, further implies $\lim_{t\to\infty} D_\Psi(w^*, w_t) = \widetilde{X}$ almost surely for a non-negative random variable $\widetilde{X}$. It is clear that $\widetilde{X} < \infty$ almost surely by Lemma E.1.

We now turn to part (a). Under the assumption $\eta_t \leq (2A)^{-1}\sigma_\Psi$ and $\eta_{t+1} \leq \eta_t$, (C.7) holds. According to (C.7) with $w = w^*$, we know

$$\sum_{k=1}^t \eta_k \big(\phi(w_k) - \phi(w^*)\big) \leq \sum_{k=1}^t \xi_k + D_\Psi(w^*, 0) + 2\sigma_\Psi^{-1}(AC_1 + B)\sum_{k=1}^t \eta_k^2, \qquad \text{(E.2)}$$

where

$$\xi_k = \eta_k \langle w^* - w_k, f'(w_k, z_k) - \mathbb{E}_{z_k}[f'(w_k, z_k)]. \rangle$$

Introduce a sequence of random variables

$$\widetilde{X}'_{t+1} = \sum_{k=1}^t \xi_k + D_\Psi(w^*, 0) + 2\sigma_\Psi^{-1}(AC_1 + B)\sum_{k=1}^\infty \eta_k^2, \quad t = 0, 1, \dots,$$

which is well defined since $\sum_{t=1}^\infty \eta_t^2 < \infty$. It is clear from $\mathbb{E}_{z_t}[\xi_t] = 0$ that

$$\mathbb{E}_{z_t}[\widetilde{X}'_{t+1}] = \sum_{k=1}^{t-1} \xi_k + \mathbb{E}_{z_t}[\xi_t] + D_\Psi(w^*, 0) + 2\sigma_\Psi^{-1}(AC_1 + B)\sum_{k=1}^\infty \eta_k^2 = \widetilde{X}'_t.$$

Furthermore, according to the definition of $w^*$ and (E.2), we know $\widetilde{X}'_t \geq 0$ for all $t \in \mathbb{N}$. Therefore, one can apply Lemma E.1 to show that $\{\widetilde{X}'_t\}_{t\in\mathbb{N}}$ converges to a non-negative variable $\widetilde{X}'$ almost surely and $\widetilde{X}' < \infty$ almost surely. This, together with (E.2) and the definition of $\widetilde{X}'_t$, implies that $\sum_{k=1}^\infty \eta_k \big(\phi(w_k) - \phi(w^*)\big) < \infty$ almost surely. This finishes the proof of part (a).

We now turn to part (b). We have shown $\lim_{t\to\infty} D_\Psi(w^*, w_t) = \widetilde{X}$ almost surely. It suffices to show $\widetilde{X} = 0$ almost surely under the condition $\sigma_\phi > 0$ and $\sum_{t=1}^\infty \eta_t = \infty$. Since $\eta_t \leq \sigma_r^{-1}$ for all $t \geq t_2$, we know

$$\frac{1 - \sigma_F \eta_t}{1 + \sigma_r \eta_t} = \frac{1 + \sigma_r \eta_t - \sigma_\phi \eta_t}{1 + \sigma_r \eta_t} \leq 1 - 2^{-1}\sigma_\phi \eta_t, \quad \forall t \geq t_2.$$

Plugging the above inequality back into (E.1) and taking expectations over both sides, we derive

$$\mathbb{E}[D_\Psi(w^*, w_{t+1})] \leq \big(1 - 2^{-1}\sigma_\phi \eta_t\big)\mathbb{E}[D_\Psi(w^*, w_t)] + \sigma_\Psi^{-1}(A\phi(w^*) + 2B)\eta_t^2, \quad \forall t \geq t_2.$$

Applying this inequality iteratively for $t = T, T-1, \dots, t_2$ yields

$$\mathbb{E}[D_\Psi(w^*, w_{T+1})] \leq \prod_{t=t_2}^T (1 - 2^{-1}\sigma_\phi \eta_t)\mathbb{E}[D_\Psi(w^*, w_{t_2})]$$

$$+ \sigma_\Psi^{-1}(A\phi(w^*) + 2B)\sum_{t=t_2}^T \eta_t^2 \prod_{k=t+1}^T (1 - 2^{-1}\sigma_\phi \eta_k), \quad \text{(E.3)}$$

where we denote $\prod_{k=t+1}^T (1 - 2^{-1}\sigma_\phi \eta_k) = 1$ for $t = T$. The first term of the above inequality can be controlled by the standard inequality $1 - a \leq \exp(-a), a > 0$ together with $\sum_{t=1}^\infty \eta_t = \infty$

$$\lim_{T\to\infty} \prod_{t=t_2}^T (1 - 2^{-1}\sigma_\phi \eta_t)\mathbb{E}[D_\Psi(w^*, w_{t_2})] \leq \lim_{T\to\infty} \prod_{t=t_2}^T \exp\big(-2^{-1}\sigma_\phi \eta_t\big)\mathbb{E}[D_\Psi(w^*, w_{t_2})]$$

$$= \lim_{T\to\infty} \exp\Big(-2^{-1}\sigma_\phi \sum_{t=t_2}^T \eta_t\Big)\mathbb{E}[D_\Psi(w^*, w_{t_2})] = 0.$$

Applying Lemma E.2 with $a = 2^{-1}\sigma_\phi$, we get $\lim_{T\to\infty}\sum_{t=t_2}^{T}\eta_t^2\prod_{k=t+1}^{T}(1 - 2^{-1}\sigma_\phi\eta_k) = 0$. Plugging the above two expressions into (E.3) implies $\lim_{T\to\infty}\mathbb{E}[D_\Psi(w^*, w_T)] = 0$. This together with Fatou's lemma shows

$$0 \le \mathbb{E}[\widetilde{X}] = \mathbb{E}\big[\lim_{T\to\infty} D_\Psi(w^*, w_T)\big] \le \liminf_{T\to\infty}\mathbb{E}[D_\Psi(w^*, w_T)] = 0,$$

from which and $\widetilde{X} \ge 0$ we know $\widetilde{X} = 0$ almost surely. This finishes the proof of part (b). The proof is complete. $\qquad\square$

## F  Proofs for Generalization Bounds

In this section, we prove generalization error bounds presented in Section 4. The following lemma is a standard probabilistic bound on the uniform deviation between empirical errors and generalization errors over a RKHS ball. In our case, we need to control the Lipschitz constants and the magnitudes for functions satisfying Assumption 1. According to (3.2) and Lemma A.4 we know $\|f'(w, z)\|_2^2 \le Af(w, z) + B$ with $A = \tilde{A}\kappa^2$ and $B = \tilde{B}\kappa^2$, where $\kappa = \sup_{x\in\mathcal{X}}\|K_x\|_2$. Recall that $f(w, z) = \ell(h_w(x), y)$.

**Lemma F.1.** *Let $R > 0$ and define $B_R = \{w \in \mathcal{W} : \|w\|_2 \le R\}$. Then, for any $\delta \in (0, 1)$, with probability at least $1 - \delta$ we have*

$$\sup_{w\in B_R}\big[\mathcal{E}(w) - \mathcal{E}_\mathbf{z}(w)\big] \le \big(C_9 R^2 + C_{10}\big)n^{-\frac{1}{2}}\log^{\frac{1}{2}}\frac{1}{\delta}, \tag{F.1}$$

*where*

$$C_9 = \kappa^2 + 2\tilde{A}^2\kappa^2 + \Big(\frac{A^2}{\sqrt{2}} + \frac{1}{2\sqrt{2}}\Big) \quad and \quad C_{10} = \Big(2\tilde{A} + \frac{A+1}{\sqrt{2}}\Big)\sup_z f(0, z) + 2\tilde{B} + \frac{B}{\sqrt{2}}.$$

*Proof.* We prove this lemma by McDiarmid's inequality (Lemma A.2). To this aim, we first show that the function $\mathbf{z} \mapsto \sup_{w\in B_R}\big[\mathcal{E}(w) - \mathcal{E}_\mathbf{z}(w)\big]$ satisfies a bounded difference property. Indeed, for any $\mathbf{z} = \{z_1, \ldots, z_{i-1}, z_i, z_{i+1}, \ldots, z_n\}$ and $\bar{\mathbf{z}} = \{z_1, \ldots, z_{i-1}, \bar{z}_i, z_{i+1}, \ldots, z_n\}$, we have

$$\Big|\sup_{w\in B_R}\big[\mathcal{E}(w) - \mathcal{E}_\mathbf{z}(w)\big] - \sup_{w\in B_R}\big[\mathcal{E}(w) - \mathcal{E}_{\bar{\mathbf{z}}}(w)\big]\Big| \le \sup_{w\in B_R}\big|\mathcal{E}_\mathbf{z}(w) - \mathcal{E}_{\bar{\mathbf{z}}}(w)\big|$$

$$\le \frac{1}{n}\sup_{w\in B_R}\big|f(w, z_i) - f(w, \bar{z}_i)\big| \le \frac{1}{n}\sup_{w\in B_R}\sup_{z\in\mathcal{Z}}f(w, z)$$

$$\le \frac{1}{n}\Big(\big(A^2 + \frac{1}{2}\big)R^2 + (A+1)\sup_z f(0, z) + B\Big),$$

where the third inequality is due to the non-negativity of $f$ and the last inequality is due to (A.5) applied to the function $w \mapsto f(w, z)$. Applying McDiarmid's inequality with increments bounded above, we derive the following inequality with probability at least $1 - \delta$

$$\sup_{w\in B_R}\big[\mathcal{E}(w) - \mathcal{E}_\mathbf{z}(w)\big] \le \mathbb{E}_\mathbf{z}\Big[\sup_{w\in B_R}\big[\mathcal{E}(w) - \mathcal{E}_\mathbf{z}(w)\big]\Big]$$

$$+ \sqrt{\frac{\log 1/\delta}{2n}}\Big(\big(A^2 + \frac{1}{2}\big)R^2 + (A+1)\sup_z f(0, z) + B\Big). \tag{F.2}$$

We now control the term $\mathbb{E}_\mathbf{z}\Big[\sup_{w\in B_R}\big[\mathcal{E}(w) - \mathcal{E}_\mathbf{z}(w)\big]\Big]$. Let $\tilde{\mathbf{z}} = \{\tilde{z}_1, \ldots, \tilde{z}_n\}$ be training examples independently drawn from $\rho$ and independent of $\mathbf{z}$. Let $\sigma_1, \ldots, \sigma_n$ be a sequence of independent Rademacher variables with $\Pr\{\sigma_i = 1\} = \Pr\{\sigma_i = -1\} = \frac{1}{2}$. By Jensen's inequality and the standard symmetrization technique, we get

$$\mathbb{E}_\mathbf{z}\Big[\sup_{w\in B_R}\big[\mathcal{E}(w) - \mathcal{E}_\mathbf{z}(w)\big]\Big] = \mathbb{E}_\mathbf{z}\Big[\sup_{w\in B_R}\big[\mathbb{E}_{\tilde{\mathbf{z}}}[\mathcal{E}_{\tilde{\mathbf{z}}}(w)] - \mathcal{E}_\mathbf{z}(w)\big]\Big]$$

$$\le \mathbb{E}_{\mathbf{z},\tilde{\mathbf{z}}}\Big[\sup_{w\in B_R}\big[\mathcal{E}_{\tilde{\mathbf{z}}}(w) - \mathcal{E}_\mathbf{z}(w)\big]\Big] = \frac{1}{n}\mathbb{E}_{\mathbf{z},\tilde{\mathbf{z}}}\Big[\sup_{w\in B_R}\sum_{i=1}^{n}\big(f(w, \tilde{z}_i) - f(w, z_i)\big)\Big]$$

$$= \frac{1}{n}\mathbb{E}_{\mathbf{z},\tilde{\mathbf{z}},\sigma}\Big[\sup_{w\in B_R}\sum_{i=1}^{n}\sigma_i\big(f(w, \tilde{z}_i) - f(w, z_i)\big)\Big] \le \frac{2}{n}\mathbb{E}_{\mathbf{z},\sigma}\Big[\sup_{w\in B_R}\sum_{i=1}^{n}\sigma_i f(w, z_i)\Big]. \tag{F.3}$$

For any $w \in B_R$, it follows from Lemma A.3 that

$$\left|\ell'(\langle w, K_x \rangle, y)\right|^2 \leq 2\tilde{A}^2 |\langle w, K_x \rangle|^2 + 2\tilde{A}\ell(0,y) + 2\tilde{B} \leq 2\tilde{A}^2 \|w\|_2^2 \|K_x\|_2^2 + 2\tilde{A}\ell(0,y) + 2\tilde{B}$$
$$\leq 2\tilde{A}^2 R^2 \kappa^2 + 2\tilde{A} \sup_y \ell(0,y) + 2\tilde{B},$$

from which we know

$$\left|\ell'(\langle w, K_x \rangle, y)\right| \leq \sqrt{2\tilde{A}^2 R^2 \kappa^2 + 2\tilde{A} \sup_y \ell(0,y) + 2\tilde{B}}, \quad \forall w \in B_R.$$

Applying Talagrand's contraction lemma [5] to the last term of (F.3) together with $f(w,z) = \ell(\langle w, K_x \rangle, y)$ and the above bound on derivative of $\ell$, we derive

$$\mathbb{E}_{\mathbf{z}}\left[\sup_{w \in B_R}\left[\mathcal{E}(w) - \mathcal{E}_{\mathbf{z}}(w)\right]\right] \leq \frac{2\sqrt{2\tilde{A}^2 R^2 \kappa^2 + 2\tilde{A} \sup_y \ell(0,y) + 2\tilde{B}}}{n} \mathbb{E}_{\mathbf{z},\sigma}\left[\sup_{w \in B_R} \sum_{i=1}^n \sigma_i \langle w, K_{x_i} \rangle\right].$$
(F.4)

According to the Schwarz's inequality and Jensen's inequality, we get

$$\mathbb{E}_\sigma\left[\sup_{w \in B_R} \sum_{i=1}^n \sigma_i \langle w, K_{x_i} \rangle\right] = \mathbb{E}_\sigma\left[\sup_{w \in B_R}\left\langle w, \sum_{i=1}^n \sigma_i K_{x_i}\right\rangle\right] \leq \mathbb{E}_\sigma\left[\sup_{w \in B_R} \|w\|_2 \sqrt{\left\|\sum_{i=1}^n \sigma_i K_{x_i}\right\|_2^2}\right]$$

$$\leq R\sqrt{\mathbb{E}_\sigma\left\langle \sum_{i=1}^n \sigma_i K_{x_i}, \sum_{i=1}^n \sigma_i K_{x_i}\right\rangle} = R\sqrt{\sum_{i=1}^n \|K_{x_i}\|_2^2} \leq R\kappa\sqrt{n}.$$

Plugging the above inequality back into (F.4), we derive

$$\mathbb{E}_{\mathbf{z}}\left[\sup_{w \in B_R}\left[\mathcal{E}(w) - \mathcal{E}_{\mathbf{z}}(w)\right]\right] \leq \frac{2R\kappa\sqrt{2\tilde{A}^2 R^2 \kappa^2 + 2\tilde{A} \sup_y \ell(0,y) + 2\tilde{B}}}{\sqrt{n}}.$$

Plugging the above inequality back into (F.2) and using $2ab \leq a^2 + b^2$ for $a, b \in \mathbb{R}$, we derive the following inequality with probability at least $1 - \delta$

$$\sup_{w \in B_R}\left[\mathcal{E}(w) - \mathcal{E}_{\mathbf{z}}(w)\right] \leq \frac{1}{\sqrt{n}}\left(R^2\kappa^2 + 2\tilde{A}^2 R^2 \kappa^2 + 2\tilde{A} \sup_y \ell(0,y) + 2\tilde{B}\right)$$

$$+ \sqrt{\frac{\log 1/\delta}{2n}}\left(\left(A^2 + \frac{1}{2}\right)R^2 + (A+1)\sup_z f(0,z) + B\right),$$

which can be written as (F.1) with the stated $C_9$ and $C_{10}$. The proof is complete. $\square$

The following lemma aims to bound $\mathcal{E}_{\mathbf{z}}(w_\lambda) - \mathcal{E}(w_\lambda)$ with $w_\lambda$ defined in (F.5). Since $w_\lambda$ is a fixed element in $\mathcal{W}$, we do not need to resort to uniform deviation arguments. Instead, we can apply a Bernstein inequality to study $\mathcal{E}_{\mathbf{z}}(w_\lambda) - \mathcal{E}(w_\lambda)$, based on the observation that Assumption 3 allows us to control the variance of $f(w_\lambda, z)$ by a linear function of $\sup_z f(w_\lambda, z)$.

**Lemma F.2.** *Let $\lambda \in (0,1]$ and define*

$$w_\lambda = \arg\min_{w \in \mathcal{W}} \mathcal{E}(w) + \lambda\|w\|_2^2. \tag{F.5}$$

*Let $\rho \in (0,1]$ and $\delta \in (0,1)$. Then, with probability at least $1 - \delta$ we have*

$$\mathcal{E}_{\mathbf{z}}(w_\lambda) - \mathcal{E}(w_\lambda) \leq \rho\bigl(c_\alpha + \mathcal{E}(h_\rho)\bigr) + (\rho n)^{-1} \sup_z f(w_\lambda, z)\log\delta^{-1}.$$

*Proof.* Let $\xi_i = f(w_\lambda, z_i), i = 1, \dots, n$. According to the definition of $w_\lambda$ and Assumption 3, we know

$$\mathcal{E}(w_\lambda) - \mathcal{E}(h_\rho) + \lambda\|w_\lambda\|_2^2 \leq c_\alpha \lambda^\alpha,$$

from which and $\lambda \leq 1$ we derive

$$\mathcal{E}(w_\lambda) \leq \mathcal{E}(h_\rho) + c_\alpha.$$

It then follows that $\xi_i - \mathbb{E}[\xi_i] \leq \sup_z f(w_\lambda, z)$ (non-negativity of $\xi_i$) and

$$\mathbb{E}\big[(\xi_i - \mathbb{E}[\xi_i])^2\big] \leq \mathbb{E}[f^2(w_\lambda, z_i)] \leq \sup_z f(w_\lambda, z)\mathbb{E}[f(w_\lambda, z)] \leq \sup_z f(w_\lambda, z)\big(c_\alpha + \mathcal{E}(h_\rho)\big).$$

Applying Part (b) of Lemma A.1 with $\xi_i = f(w_\lambda, z_i)$ and the above bounds on variances and magnitudes, we derive the following inequality with probability at least $1 - \delta$

$$\mathcal{E}_{\mathbf{z}}(w_\lambda) - \mathcal{E}(w_\lambda) = \frac{1}{n}\sum_{i=1}^n \xi_i - \mathbb{E}[\xi] \leq \frac{\rho n \sup_z f(w_\lambda, z)\big(c_\alpha + \mathcal{E}(h_\rho)\big)}{n \sup_z f(w_\lambda, z)} + \frac{\sup_z f(w_\lambda, z)\log\frac{1}{\delta}}{\rho n}.$$

The stated inequality then follows directly. The proof is complete. $\qquad\square$

We are now in a position to prove Theorem 10. Our basic idea is to use the decomposition (F.6) with $w_\lambda$ and $\lambda$ proportional to $n^{-\frac{\alpha}{1+\alpha}}$. The term $\mathcal{E}_{\mathbf{z}}(\bar{w}_T^{(1)}) - \mathcal{E}_{\mathbf{z}}(w_\lambda)$ is the computational error related to the optimization process. Both $\mathcal{E}(\bar{w}_T^{(1)}) - \mathcal{E}_{\mathbf{z}}(\bar{w}_T^{(1)})$ and $\mathcal{E}_{\mathbf{z}}(w_\lambda) - \mathcal{E}(w_\lambda)$ are estimation errors related to the sampling process. The term $\mathcal{E}(w_\lambda) - \mathcal{E}(h_\rho)$ is the approximation error. In the following, we apply Lemma F.1 and Lemma F.2 to control estimation errors, Theorem 4 to control the computational error and Assumption 3 to control the approximation error. Here we use three tricks to get almost optimal generalization error bounds. First, we show that $\|\bar{w}_T^{(1)}\|_2^2$ grows as a logarithmic function of $T$, which allows us to get $\mathcal{E}(\bar{w}_T^{(1)}) - \mathcal{E}_{\mathbf{z}}(\bar{w}_T^{(1)}) = O(n^{-\frac{1}{2}}\log T)$ (we omit the dependency on $1/\delta$ for brevity). Second, in the analysis of $\mathcal{E}_{\mathbf{z}}(w_\lambda) - \mathcal{E}(w_\lambda)$, we show the variance of $f(w_\lambda, z)$ grows as a linear function of $\sup_z f(w_\lambda, z)$ instead of a quadratic function of $\sup_z f(w_\lambda, z)$ by exploiting Assumption 3, which allows us to get a bound with a mild dependency on $\|w_\lambda\|_2^2$. As a comparison, if we use $\|w_\lambda\|_2^2 = O(\lambda^{\alpha-1})$ due to Assumption 3 and the Azuma-Hoeffding inequality we will get $\mathcal{E}_{\mathbf{z}}(w_\lambda) - \mathcal{E}(w_\lambda) = O(\lambda^{\alpha-1}n^{-\frac{1}{2}})$, which is suboptimal since $\lambda$ is chosen to be very small to trade the estimation, computational and approximation errors. Indeed, if one plug $\mathcal{E}_{\mathbf{z}}(w_\lambda) - \mathcal{E}(w_\lambda) = O(\lambda^{\alpha-1}n^{-\frac{1}{2}})$ into (F.6), one can only derive the suboptimal bound $\mathcal{E}(\bar{w}_T^{(1)}) - \mathcal{E}(h_\rho) = O(n^{-\frac{\alpha}{2}}\log^{\frac{3}{2}} T)$ worse than $O(n^{-\frac{\alpha}{1+\alpha}}\log^{\frac{3}{2}} T)$ in Theorem 10. The third trick is to choose $w_\lambda$ with an appropriate $\lambda$ in (F.6) to fully exploit Assumption 3.

*Proof of Theorem 10.* Let $\lambda, \rho \in (0, 1]$ be real numbers to be fixed later and $w = w_\lambda$ defined by (F.5). We use the following error decomposition w.r.t. $w_\lambda$ to study the excess generalization error $\mathcal{E}(\bar{w}_T^{(1)}) - \mathcal{E}(h_\rho)$

$$\begin{aligned}
\mathcal{E}(\bar{w}_T^{(1)}) - \mathcal{E}(h_\rho) = \big(\mathcal{E}(\bar{w}_T^{(1)}) - \mathcal{E}_{\mathbf{z}}(\bar{w}_T^{(1)})\big) &+ \big(\mathcal{E}_{\mathbf{z}}(\bar{w}_T^{(1)}) - \mathcal{E}_{\mathbf{z}}(w_\lambda)\big) \\
&+ \big(\mathcal{E}_{\mathbf{z}}(w_\lambda) - \mathcal{E}(w_\lambda)\big) + \big(\mathcal{E}(w_\lambda) - \mathcal{E}(h_\rho)\big).
\end{aligned} \quad \text{(F.6)}$$

It is clear that (4.1) is a specific instantiation of (2.2) with $f(w, z) = \ell(\langle w, K_x\rangle, y)$, $\Psi(w) = \frac{1}{2}\|w\|_2^2$, $r(w) = 0$ and $\tilde{\rho}$ being the uniform distribution over $\{z_1, \ldots, z_n\}$. During the iteration of (4.1), the training sample $\mathbf{z} = \{z_1, \ldots, z_n\}$ is fixed and the randomness comes from the index sequence $\{j_t\}_{t\in\mathbb{N}}$. Since $j_t$ is drawn from a uniform distribution over $\{1, \ldots, n\}$, the objective function minimized by the SGD scheme (4.1) is the empirical error $\phi(w) = \mathbb{E}_{j_t}[f(w, z_{j_t})] = \mathcal{E}_{\mathbf{z}}(w)$. An application of Theorem 4 to the SGD scheme (4.1) with $w = w_\lambda$ then gives the following inequality with probability $1 - \delta/4$

$$\mathcal{E}_{\mathbf{z}}(\bar{w}_T^{(1)}) - \mathcal{E}_{\mathbf{z}}(w_\lambda) \leq \Big(\sum_{t=1}^T \eta_t\Big)^{-1}\big(C_3\|w_\lambda\|_2^2 + C_4\big)\log^{\frac{3}{2}}\frac{8T}{\delta}. \quad \text{(F.7)}$$

We can apply Lemma F.2 to derive the following inequality with probability at least $1 - \delta/4$

$$\begin{aligned}
\mathcal{E}_{\mathbf{z}}(w_\lambda) - \mathcal{E}(w_\lambda) &\leq \rho\big(c_\alpha + \mathcal{E}(h_\rho)\big) + (\rho n)^{-1}\sup_z f(w_\lambda, z)\log\frac{4}{\delta} \\
&\leq \rho\big(c_\alpha + \mathcal{E}(h_\rho)\big) + (\rho n)^{-1}\Big(\Big(A^2 + \frac{1}{2}\Big)\|w_\lambda\|_2^2 + (A+1)\sup_z f(0, z) + B\Big)\log\frac{4}{\delta},
\end{aligned}$$
$$\text{(F.8)}$$

where the last inequality is due to Lemma A.3.

According to Theorem 3, with probability at least $1 - \delta/4$ we have $\max_{1 \leq t \leq T} \|w_t\|_2 \leq \sqrt{C_2 \log \frac{4T}{\delta}}$, from which and the convexity of norm we derive the following inequality with probability $1 - \delta/4$

$$\|\bar{w}_T^{(1)}\|_2 \leq \sqrt{C_2 \log \frac{4T}{\delta}}. \tag{F.9}$$

Furthermore, an application of Lemma F.1 with $\widetilde{R} = \sqrt{C_2 \log \frac{4T}{\delta}}$ shows the following inequality with probability $1 - \delta/4$

$$\sup_{w \in B_{\widetilde{R}}} \left[ \mathcal{E}(w) - \mathcal{E}_{\mathbf{z}}(w) \right] \leq \left( C_9 C_2 \log \frac{4T}{\delta} + C_{10} \right) n^{-\frac{1}{2}} \log^{\frac{1}{2}} \frac{4}{\delta}.$$

Combining the above inequality and (F.9) together, we derive the following inequality with probability $1 - \delta/2$

$$\left[ \mathcal{E}(\bar{w}_T^{(1)}) - \mathcal{E}_{\mathbf{z}}(\bar{w}_T^{(1)}) \right] \leq \left( C_9 C_2 + C_{10} \right) n^{-\frac{1}{2}} \log^{\frac{3}{2}} \frac{4T}{\delta}. \tag{F.10}$$

Plugging (F.7), (F.8) and (F.10) into (F.6), we derive the following inequality with probability at least $1 - \delta$

$$\mathcal{E}(\bar{w}_T^{(1)}) - \mathcal{E}(h_\rho) \leq \mathcal{E}(w_\lambda) - \mathcal{E}(h_\rho) + \|w_\lambda\|_2^2 \left( C_3 \left( \sum_{t=1}^{T} \eta_t \right)^{-1} + (\rho n)^{-1} \left( A^2 + 2^{-1} \right) \right) \log^{\frac{3}{2}} \frac{8T}{\delta}$$

$$+ C_4 \left( \sum_{t=1}^{T} \eta_t \right)^{-1} \log^{\frac{3}{2}} \frac{8T}{\delta} + \left( C_9 C_2 + C_{10} \right) n^{-\frac{1}{2}} \log^{\frac{3}{2}} \frac{4T}{\delta}$$

$$+ \rho \left( c_\alpha + \mathcal{E}(h_\rho) \right) + (\rho n)^{-1} \left( (A+1) \sup_z f(0, z) + B \right) \log \frac{4}{\delta}.$$

We choose $\lambda = \max \left\{ \left( \sum_{t=1}^{T} \eta_t \right)^{-1}, (\rho n)^{-1} \right\}$ in the above inequality and derive the following inequality with probability $1 - \delta$

$$\mathcal{E}(\bar{w}_T^{(1)}) - \mathcal{E}(h_\rho) \leq \left( C_3 + A^2 + 2^{-1} \right) D \left( \max \left\{ \left( \sum_{t=1}^{T} \eta_t \right)^{-1}, (\rho n)^{-1} \right\} \right) \log^{\frac{3}{2}} \frac{8T}{\delta} + \left( C_4 \left( \sum_{t=1}^{T} \eta_t \right)^{-1} + \right.$$

$$\left. \left( C_9 C_2 + C_{10} \right) n^{-\frac{1}{2}} \right) \log^{\frac{3}{2}} \frac{8T}{\delta} + \rho \left( c_\alpha + \mathcal{E}(h_\rho) \right) + (\rho n)^{-1} \left( (A+1) \sup_z f(0, z) + B \right) \log \frac{4}{\delta},$$

where in the first inequality we have used $C_3 + A^2 + 2^{-1} \geq 1$ and

$$\mathcal{E}(w_\lambda) - \mathcal{E}(h_\rho) + \|w_\lambda\|_2^2 \left( C_3 \left( \sum_{t=1}^{T} \eta_t \right)^{-1} + (\rho n)^{-1} \left( A^2 + 2^{-1} \right) \right) \log^{\frac{3}{2}} \frac{8T}{\delta}$$

$$\leq \left( C_3 + A^2 + 2^{-1} \right) \left( \mathcal{E}(w_\lambda) - \mathcal{E}(h_\rho) + \lambda \|w_\lambda\|_2^2 \right) \log^{\frac{3}{2}} \frac{8T}{\delta} = \left( C_3 + A^2 + 2^{-1} \right) D(\lambda) \log^{\frac{3}{2}} \frac{8T}{\delta}.$$

Since the above inequality holds for any $\rho \in (0, 1]$, we can take $\rho = n^{-\frac{\alpha}{1+\alpha}}$ to derive the following inequality with probability at least $1 - \delta$

$$\mathcal{E}(\bar{w}_T^{(1)}) - \mathcal{E}(h_\rho) \leq c_\alpha \left( C_3 + A^2 + 2^{-1} \right) \max \left\{ \left( \sum_{t=1}^{T} \eta_t \right)^{-\alpha}, n^{-\frac{\alpha}{1+\alpha}} \right\} \log^{\frac{3}{2}} \frac{8T}{\delta} + \left( C_4 \left( \sum_{t=1}^{T} \eta_t \right)^{-1} + \right.$$

$$\left. \left( C_9 C_2 + C_{10} \right) n^{-\frac{1}{2}} \right) \log^{\frac{3}{2}} \frac{8T}{\delta} + n^{-\frac{\alpha}{1+\alpha}} \left( c_\alpha + \mathcal{E}(h_\rho) + (A+1) \sup_z f(0, z) + B \right) \log \frac{4}{\delta},$$

from which it follows directly the stated inequality (4.2) with $C_5$ defined by

$$C_5 = c_\alpha (C_3 + A^2 + 2^{-1}) + C_4 + C_9 C_2 + C_{10} + c_\alpha + \mathcal{E}(h_\rho) + (A+1) \sup_z f(0, z) + B.$$

It is clear both $\rho$ and $\lambda$ defined above satisfy $\rho, \lambda \in (0, 1]$. The proof is complete. $\qquad \square$

| Dataset | No. of Training Examples | No. of Test Examples | No. of Attributes | Source |
|---------|--------------------------|----------------------|-------------------|--------|
| ADULT | $32,561$ | $16,281$ | $123$ | [7] |
| GISETTE | $6,000$ | $1,000$ | $5,000$ | [3] |
| IJCNN1 | $49,990$ | $91,701$ | $22$ | [8] |
| MUSHROOMS | $4,062$ | $4,062$ | $112$ | [1] |
| PHISHING | $5,527$ | $5,528$ | $68$ | [1] |
| SPLICE | $1,000$ | $2,175$ | $60$ | [1] |

Table G.1: Description of datasets used in the experiments.

## G  Additional Information on Simulation

We present a detailed description of datasets, used in Section 6, in Table G.1.