[Reviews · NeurIPS 2018]

Reviewer 1



This paper studied stochastic composite mirror descent, for both convex and strongly convex objectives, and established high probability bounds. The results are stronger than existing ones from the perspective of removing smoothness assumptions or improving the rates, removing the bounded subgradient assumption, and obtaining high probability bounds rather than expectation. The authors also studied the scaling of the estimation error with respect to the number of iterations of the SGD. It seems that the authors claimed that the introduction of Assumption 1 is novel, as lines 113-117 said, and also there were some interesting martingale arguments. The results are clearly stated. The reviewer finds this paper non-trivial, but it is not clear whether the contributions are significant. In particular, the rates in this paper have essentially been obtained in the past, and it is the reviewer's understanding that a more careful martingale analysis and Assumption 1 help weaken the previous assumptions. The reviewer thinks the paper has some incremental value.

Reviewer 2



* summary: In this paper, the authors studied the optimization and generalization of the stochastic composite mirror descent (SCMD) method for machine learning problems. In the convergence analysis, a common assumption, i.e., the boundedness of the gradient vector, was not assumed. Then, the generalization ability of the kernel-based SGD method was established. The generalization bound revealed the fact that the estimation errors in the generalization error will never essentially dominate the approximation and computation errors. As a result, one can run the SGD with a sufficient number of iterations with little overfitting if step sizes are square-summable. Some numerical experiments were conducted to confirm these theoretical findings. * comments: The authors mentioned their motivation clearly, and they could successfully remove the common boundedness assumption of the gradient vector from the theoretical analysis of the SGD. Some related works were discussed in detail. Numerical experiments agreed to the theoretical findings. - It is not clear whether removing the bounded gradient assumption is the key to obtain the assertion in line 309. How does the following finding in line 309 relate to the bounded gradient assumption? > the estimation errors in the generalization error will never essentially dominate the approximation > and computation errors and one can run SGD with a sufficient number of iterations with little > overfitting if step sizes are square-summable. - line 194: Is Psi(w) equal to 1/2 |w|^2? - Assumption 3: Showing a typical value of alpha for several problem setups would be helpful for readers. - Should the kernel function be universal to agree to Assumption 3? (The universal kernel is defined in [Steinwart and Christmann, Support Vector Machines, 2008]). Adding a supplementary comment about the relation between Assumption 3 and some properties of kernel functions would be nice.

Reviewer 3



The paper focused on the stochastic composite mirror descent algorithms which update model parameters sequentially with cheap per-iteration computation, making them amenable for large-scale streaming data analysis. The setting is quite general including general convex objectives and penalty terms, which could be just convex or strongly convex. The main contribution of the paper is the almost optimal rates with high probability which match the minimax low rates for stochastic first-order optimization algorithms. The techniques are quite delicate and novel which are summarized in lemma 1 and Theorem 3. Another technique is a weighted summation which originates from [29]. In summary, the paper presents novel theoretical convergence rates for SCMD which are almost optimal, up to a logarithmic term. These comprehensive results are in the form of high probabilities which are non-trivial extension and refinement of existing literature. The paper is well presented and Section 5 on related work well summarized the main contribution of the work in a detailed comparison with existing work. Minor comments: 1. The discussion about Theorem 4 and Corollary 5 in lines 136-140 should be moved after the statement of theorem 4 and corollary 5. 2. In line 198, Assumption 3 is mentioned before it is introduced. 3. Personally, I do not see what is the main purpose of Section 6 (simulations). The main meat of the paper is novel theory which is quite satisfactory to me. I read the authors' feedback. I am satisfied with their response.

Reviewer 4



The paper presents a high probability analysis of SGD for non-smooth losses, with weak assumptions on domain and gradients. It presents many different results, some of them are very interesting while some others are nice to have but not surprising at all. In details: - Section 3.1 presents high probability convergence rates for convex smooth functions. The core trick is to show that the iterates of SGD are (almost) bounded with high probability for square summable stepsizes. This result is not surprising at all: classic results on SGD already proved convergence with probability 1 even under weaker assumptions (i.e. non-convex functions with well-behaved gradients, see for example [6]) and through the same reasoning. The high probability bounds are nice to have, but nothing earth-shattering. - Section 3.2 has similar results for strongly convex functions. These are even less suprising: changing the proof of [17] to make it work for smooth losses is a simple excercise that many of us have already done on their own. The high probability bounds for strongly objective functions are well-known, even in the composite mirror descent algorithm. Also, despite the claims in lines 274-275, removing the boundedness assumption on the gradients to obtain high probability bounds is trivial: It is enough to add a projection step into the ball where the optimal solution lives, that is always known using f(0, z_t)+r(0) and the strong convexity constant that you assumed to know. Indeed, this is how the Pegasos software was analyzed and implemented, [Shalev-Shwartz et al. ICML'07] (not cited). The projection step results in empirical faster convergence, better resistance to mispecified stepsizes, and even with better constants in the convergence rate. So, not using a projection step is actually a very bad idea. Overall, this is also a nice to have result, but not too strong. - Section 4 presents the application of these results to prove generalization bounds for a multiple pass procedure that minimizes the unregularized empirical error. The obtained convergence rate matches the one obtained in Theorem 3 in [Orabona, NIPS'14] (not cited) with worst constants, yet here the authors don't assume bounded gradients and run multiple passes. I found this result very interesting, probably the most important one in the paper. As said, a way to achieve this rate without tuning stepsizes was known, but the fact that SGD achieves this rate after multiple passes, with a very weak dependency on the number of passes, is somehow suprising to me and it improves the result obtained with the stability analysis of SGD. Overall, in my opinion the result presented in Section 4 warrants the publication of this paper. Also, it would be nice to see less overselling about the bounded gradients business.